# UniMoT: Unified Molecule-Text Language Model with Discrete Token Representation

## Abstract

The remarkable success of Large Language Models (LLMs) across diverse tasks has driven the research community to extend their capabilities to molecular applications, leading to the development of molecular LLMs. However, most molecular LLMs employ adapter-based architectures that do not treat molecule and text modalities equally and lack a supervision signal for the molecule modality. To address these issues, we introduce **UniMoT**, a unified molecule-text LLM adopting a tokenizer-based architecture that expands the vocabulary of LLM with molecule tokens. Specifically, we introduce a Vector Quantization-driven tokenizer that incorporates a Q-Former to bridge the modality gap between molecule and text. This tokenizer transforms molecules into sequences of molecule tokens with causal dependency, encapsulating high-level molecular and textual information. Equipped with this tokenizer, UniMoT can unify molecule and text modalities under a shared token representation and an autoregressive training paradigm, enabling it to interpret molecules as a foreign language and generate them as text. Following a four-stage training scheme, UniMoT emerges as a multi-modal generalist capable of performing both molecule-to-text and text-to-molecule tasks. Extensive experiments demonstrate that UniMoT achieves state-of-the-art performance across a wide range of molecule comprehension and generation tasks.

## 1 Introduction

The incredible capabilities of Large Language Models (LLMs) [5, 44] have led to their widespread use as versatile tools for completing diverse real-world tasks. This success has sparked interest in Multi-modal LLMs [59, 52], which aim to enhance LLMs by enabling them to process multi-modal inputs and outputs. Prior research efforts [26, 41, 12, 6, 33, 35, 25] have focused on adapting LLMs to molecular tasks, resulting in the development of molecular LLMs. These molecular LLMs can analyze molecule structures [35, 33, 6], address drug-related inquiries [26, 41], assist in synthesis and retrosynthesis planning [12], support drug design [12], and more.

Prevalent molecular LLMs commonly employ adapter-based architectures, adopting either a linear projection [26, 41, 6] or a Q-Former [33, 25] as an adapter to translate molecule features into the semantic space of LLM, as illustrated in Figure 1a and Figure 1b. Despite demonstrating initial capabilities in molecular comprehension and yielding promising results in molecule-to-text generation tasks, they still lack molecule generation abilities. The critical issue within these methods is their unequal treatment of molecules and text, resulting in a lack of supervision for the molecule modality. This limitation significantly constrains model capacity and effectiveness. Due to limitations imposed by the training paradigm, they are unable to perform text-to-molecule generation tasks.

Discretizing continuous molecule features into discrete molecule tokens offers a promising solution for conducting both molecule-to-text and text-to-molecule generation tasks. By treating tokens from

Submitted to 38th Conference on Neural Information Processing Systems (NeurIPS 2024). Do not distribute.

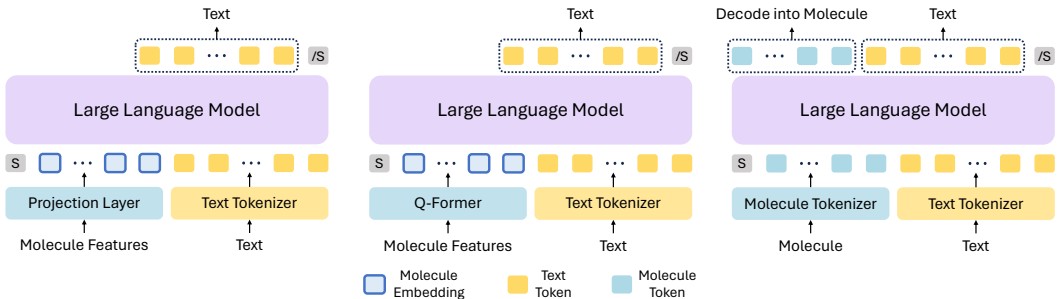

(a) Projection-Based Architecture.    (b) Q-Former-Based Architecture.    (c) Tokenizer-Based Architecture.

Figure 1: Comparisons among different molecular LLMs. 1a and 1b are adapter-based architectures that do not treat molecule and text modalities equally and lack a supervision signal for the molecule modality. 1c is our proposed tokenizer-based architecture, where molecules are presented in the same discrete token representation as that of text. Molecules and text can be optimized under a unified next-token-prediction objective.

different modalities equally, we can predict the next molecule or text token in an autoregressive manner. However, directly discretizing molecule features poses several challenges: (i) This approach results in long sequences, with lengths equivalent to the number of atoms in a batch. LLMs typically experience a quadratic increase in computational complexity with sequence length [46]. (ii) Molecule tokens derived from molecule features lack left-to-right causal dependency, which conflicts with the unidirectional attention mechanism in LLMs. (iii) Molecule features lack textual information, hindering effective molecule-text interactions and alignment.

To this end, we present **UniMoT**, a unified molecule-text LLM that adopts a tokenizer-based architecture, integrating molecule comprehension and generation, as depicted in Figure 1c. A pivotal aspect of UniMoT's architecture is the molecule tokenizer for transforming molecules into molecule tokens. We introduce a Vector Quantization-driven [45] tokenizer, incorporating a Q-Former [23] to bridge the modality gap between molecule and text. Specifically, we incorporate causal masks for the queries, enabling the Causal Q-Former to generate a causal sequence of query embeddings compatible with the unidirectional attention in LLMs. The sequence of query embeddings is subsequently quantized into a sequence of molecule tokens using a learnable codebook. The molecule tokens encapsulate high-level molecular and textual information, which are then aligned with the latent space of a generative model via an MLP adapter, enabling the generation of desired molecules.

Pretrained LLMs can integrate the molecule tokenizer by treating molecule tokens as new words and constructing a molecule vocabulary through mapping the learned codebook. We adopt the unified discrete token representation for molecules and text, coupled with the unified next-token-prediction training paradigm of LLM. This unification of representation and training paradigm enhances LLMs' ability to understand molecule-text interactions and alignment. UniMoT interprets molecules akin to understanding a foreign language, and generates them as if they were text. Following a four-stage training scheme, UniMoT serves as a multi-modal generalist capable of performing both molecule comprehension and generation tasks.

Our contributions can be summarized as follows:

- We introduce a molecule tokenizer specifically designed for LLMs, enabling the tokenization of molecules into short sequences of molecule tokens with causal dependency. These tokens encapsulate high-level molecular and textual information and can be decoded into desired molecules during inference.

- We present UniMoT, a unified molecule-text LLM that adopts a tokenizer-based architecture instead of traditional adapter-based architectures. UniMoT unifies the modalities of molecule and text under a shared token representation and an autoregressive training paradigm.

- UniMoT exhibits remarkable capabilities in multi-modal comprehension and generation. Extensive experiments demonstrate that UniMoT achieves state-of-the-art performance across a wide spectrum of molecule comprehension tasks and molecule generation tasks.

## 2 Related Works

**Molecular Large Language Models.**    The recent emergence of Vision Large Language Models (VLLMs) [24, 23, 28] has catalyzed advancements in Molecular LLMs, which encompass both single modality and multi-modality approaches. In the single modality domain, researchers are exploring diverse molecule representations, such as 1D sequences like SMILES strings [47, 8, 17], 2D molecule graphs [15, 56], 3D geometric conformations [56, 32], and textual information from the literature [43, 2, 21]. In the multiple modalities domain, various innovative approaches are being employed. MolT5 [11], a T5-based [38] model, is designed for SMILES-to-text and text-to-SMILES translations. Other works, such as MoMu [39], MoleculeSTM [31], MolFM [34], and GIT-Mol [29], leverage cross-modal contrastive learning to align the representation spaces of molecules and text. Additionally, some studies use multi-modal learning architectures to develop molecular LLMs, which often adopt adapter-based architectures. For instance, InstructMol [6], GraphGPT [41], and DrugChat [26] employ a simple projection layer to map molecule features to LLM's input space. MolCA [33] and 3D-MoLM [25] utilize a Q-Former [23] to bridge the modality gap between molecules and text. However, these methods do not treat molecule and text modalities equally and lack a supervision signal for the molecule modality, limiting model capacity and effectiveness.

**Vector Quantization.**    Vector Quantization (VQ) [13] is a widely used technique in generative models. VQ-VAE [45] converts an image into a set of discrete codes within a learnable discrete latent space by learning to reconstruct the original image. VQ-GAN [57] enhances the generation quality by leveraging adversarial and perceptual objectives. In the context of molecules, VQ has been effectively applied to quantize molecule representations. For example, DGAE [4] introduces a VQ model specifically for molecular graphs, where molecular graphs are encoded into discrete latent codes. Mole-BERT [54] uses VQ to rethink the pre-training of GNNs for molecular tasks. IMoLD [60] proposes using VQ to enhance invariant molecule representations, and VQSynergy [51] demonstrates the use of VQ for drug discovery.

## 3 Method

Our objective is to leverage the reasoning and generation capabilities of LLMs to enhance the comprehension and generation of molecule and text data. To achieve this, we focus on representing these modalities uniformly within the token representation, utilizing the next-token-prediction training paradigm of LLMs. As illustrated in Figure 2, we introduce a molecule tokenizer (Section 3.1) designed to transform molecules into molecule tokens by learning to reconstruct the input molecule. The molecule sequence can then be concatenated with the text sequence to form a multi-modal sequence, which is subsequently fed into an LLM for autoregressive pretraining (Section 3.2), as illustrated in Figure 3. The LLM vocabulary is expanded with molecule codes mapped from the learned codebook. We introduce a four-stage training scheme for UniMoT (Section 3.3) comprising Causal Q-Former pretraining, molecule tokenizer pretraining, unified molecule-text pretraining, and task-specific instruction tuning. UniMoT is capable of performing both molecular comprehension and generation tasks following the training scheme.

### 3.1 Molecule Tokenizer for LLMs

**Molecule encoder.**    We represent the structural information of a molecule as a graph, denoted by $\mathcal{G} = (\mathcal{V}, \mathcal{E})$, where $\mathcal{V}$ is the set of atoms and $|\mathcal{V}| = N$ is the number of atoms. The task of the molecule encoder is to extract node representations that are context-aware and encompass diverse local neighborhood structural information. By employing a molecule encoder, we obtain molecule features $\mathbf{X} \in \mathbb{R}^{N \times F}$, where each atom representation contains context-aware structural information.

**Causal Q-Former.**    We employ a Q-Former model introduced by BLIP-2 [23] to generate query embeddings $\mathbf{Z} = \{z_i\}_{i=1}^{M} \in \mathbb{R}^{M \times d}$ containing high-level molecular and textual information, where $M$ represents the number of queries and $d$ denotes the dimension of query embeddings. Specifically, we incorporate causal masks into the queries, ensuring that they only interact with preceding queries. This ensures the sequence of query embeddings maintains a causal dependency, aligning with the requirements of LLMs operating on text sequence. Details regarding the Causal Q-Former can be found in Appendix A.

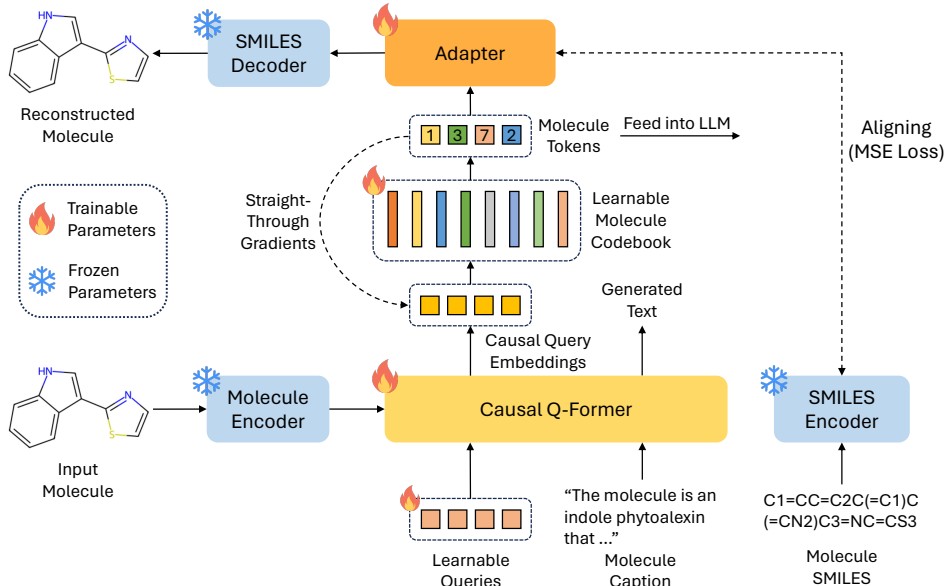

Figure 2: Illustration of our proposed molecule tokenizer. The tokenizer generates discrete molecule tokens, which can be fed into LLMs for downstream tasks. The generated molecule tokens can be decoded into molecules using the adapter and the SMILES decoder during inference.

**Vector Quantization.** The Causal Q-Former converts molecule and text features into a causal sequence of query embeddings. Subsequently, we aim to quantize these query embeddings into molecule tokens using a variant of VQ-VAE [45]. These discrete molecule tokens can then be integrated with text tokens to form a multi-modal sequence suitable for feeding into LLMs. The causal sequence of query embeddings $\{z_i\}_{i=1}^M$ is quantized into a causal sequence of molecule tokens $\{s_i\}_{i=1}^M$ by identifying the closest neighbor in a learnable codebook $\mathcal{C} = \{c_i\}_{i=1}^K$, where $K$ represents the size of the codebook. The codebook is randomly initialized and optimized during pretraining. Specifically, token $s_i$ is determined as follows:

$$s_i = \operatorname{argmin}_{j \in \{1, \cdots, K\}} \|z_i - c_j\|_2, \quad \text{for} \quad i = 1, 2, \cdots, M. \tag{1}$$

Intuitively, the query embedding $z_i$ is quantized to the closest neighbor $c_{s_i}$ in the codebook. As the vector quantization process is non-differentiable, we adopt the straight-through estimator [3] to train the Causal Q-Former by copying the gradient from the molecule tokens to the query embeddings, as shown in Figure 2. The resulting embeddings of molecule tokens, denoted as $\mathbf{C} = \{c_{s_i}\}_{i=1}^M$, are subsequently utilized for reconstructing molecules.

**Molecule Reconstruction.** An adapter needs to be trained to align the discrete latent space of molecule tokens with the continuous latent space of a molecular generative model for molecule reconstruction. The embeddings of molecule tokens $\mathbf{C}$ can be aligned with the latent space of the generative model via an MLP adapter $\psi$, represented as $\mathbf{X}_R = \psi(\mathbf{C})$, where $\mathbf{X}_R$ denotes the embeddings for reconstruction. Subsequently, we can reconstruct the molecule from $\mathbf{X}_R$ using the pretrained SMILES decoder To achieve alignment, we minimize the Mean Squared Error (MSE) loss between $\mathbf{X}_R$ and the SMILES [50] embeddings $\mathbf{X}_S$ produced by the pretrained SMILES encoder. The training loss of the tokenizer is expressed as follows:

$$\mathcal{L}_{\text{Tokenizer}} = \|\mathbf{X}_R - \mathbf{X}_S\|_2^2 + \frac{1}{M}\sum_{i=1}^M \|\text{sg}\,[z_i] - c_{s_i}\|_2^2 + \frac{\beta}{M}\sum_{i=1}^M \|\text{sg}\,[c_{s_i}] - z_i\|_2^2. \tag{2}$$

Here, the first term represents the alignment loss, the second term is a codebook loss aimed at updating the codebook embeddings, and the third term is a commitment loss that encourages the query embedding to stay close to the chosen codebook embedding. $\text{sg}[\cdot]$ denotes the stop-gradient operator, and the hyperparameter $\beta$ is set to 0.25.

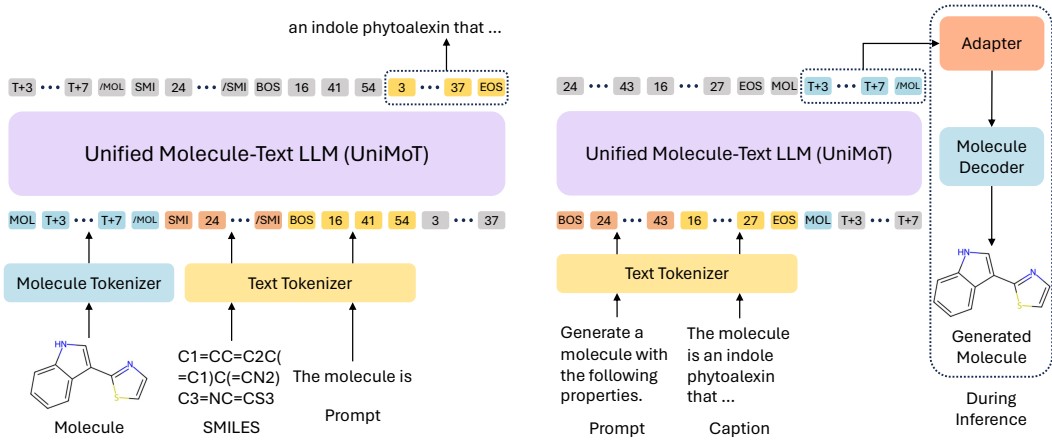

| (a) Molecule-to-Text Autoregression. | (b) Text-to-Molecule Autoregression. |

Figure 3: Illustration of the multi-modal autoregressive pretraining on molecule-text datasets. Uni-MoT excels in multi-modal comprehension and generation tasks, enabled by the unified LM objective. $T$ represents the size of the text vocabulary.

## 3.2 Unified Molecule-Text Language Model

**Expanding Vocabulary.** Employing the molecule tokenizer, a molecule can be tokenized into a molecule sequence $\{s_i\}_{i=1}^{M}$ with causal dependency. The molecule sequence can be concatenated with the text sequence to form a multi-modal sequence $\{u_i\}_{i=1}^{L}$, where $L$ is the length of the multi-modal sequence. To facilitate the representation of the multi-modal sequence, we construct the molecule vocabulary $\mathcal{V}^m = \{\boldsymbol{v}_i^m\}_{i=1}^{K}$, which maintains the order of the molecule codebook $\mathcal{C} = \{\boldsymbol{c}_i\}_{i=1}^{K}$. Additionally, $\mathcal{V}^m$ includes several special tokens such as boundary indicators, e.g., [MOL] and [/MOL], to mark the beginning and end of the molecule sequence. Next, we merge the original text vocabulary $\mathcal{V}^t = \{\boldsymbol{v}_i^t\}_{i=1}^{T}$ with the molecule vocabulary $\mathcal{V}^m$. The unified molecule-text vocabulary $\mathcal{V} = \{\mathcal{V}^m, \mathcal{V}^t\}$ facilitates joint learning from molecules and text under a unified next-token-prediction objective. As the vocabulary is expanded, the corresponding embeddings and prediction layers also need to be extended, with the newly introduced parameters initialized randomly.

**Unified Molecule-text Modeling.** The multi-modal sequence $\{u_i\}_{i=1}^{L}$ is fed into the pretrained LLM for performing multi-modal autoregression. UniMoT adopts the general Language Modeling (LM) objective to directly maximize the log-likelihood of the data distribution:

$$\mathcal{L}_{\text{LM}} = -\sum_{u \in \mathcal{D}} \sum_{i \in \mathcal{I}} \log p\left(u_i \mid u_1, \cdots, u_{i-1}; \theta\right), \qquad (3)$$

where $\mathcal{D}$ represents the dataset, $\mathcal{I}$ represents the set of indices of the generation target, and $\theta$ denotes the parameters of the LLM. The unification of representation and training paradigm for molecules and text enhances the abilities of LLMs to understand molecule-text interactions and alignment. UniMoT can interpret molecules similar to understanding a foreign language, and generate them as if they were text. We conduct autoregressive pretraining on molecule-to-text and text-to-molecule tasks to enhance the molecule comprehension and generation capabilities.

**Molecule-to-Text Autoregression.** While structural information is embedded in molecule features and captured by the molecule tokens through the tokenizer, we also aim to incorporate sequential information of molecules for better comprehension. Therefore, we concatenate the molecule sequence $\{s_i\}_{i=1}^{M}$ with the SMILES [50] sequence and a prompt to form the multi-modal input sequence $\{u_i\}_{i=1}^{L}$, as illustrated in Figure 3a. The text sequence of the corresponding molecule caption is used as the generation target.

**Text-to-Molecule Autoregression.** For molecule generation, a prompt and the molecule caption are concatenated, with a [MOL] token appended to signify the beginning of the molecule sequence, as illustrated in Figure 3b. The molecule sequence $\{s_i\}_{i=1}^{M}$ produced by the tokenizer is used as the

generation target. During inference, given a prompt and the molecule caption, the output molecule
sequence can be decoded into the desired molecule by the pretrained adapter and SMILES decoder.

### 3.3 Training Strategy

The training strategy for UniMoT is structured across four stages. Stage-1 focuses on Causal Q-Former pretraining with tailored objectives. In Stage-2, the molecule tokenizer is optimized using the frozen encoders and decoder. Stage-3 integrates the tokenizer with a language model for multi-modal comprehension and generation. Finally, Stage-4 fine-tunes UniMoT for specific tasks, aligning it with human instructions and optimizing performance for various molecular applications. More details regarding the training process can be found in Appendix C.

**Stage-1: Causal Q-Former Pretraining.**  We connect the molecule encoder and Causal Q-Former, leveraging the pretrained MoleculeSTM molecule encoder [31]. The molecule encoder remains frozen while only the Causal Q-Former is updated. Both queries and text inputs are used, while only queries serve as input in subsequent stages. In our experiments, we utilize 16 queries. We employ three tailored objectives MTC, MTM, and MTG for the pretraining of the Causal Q-Former, as detailed in Appendix A.

**Stage-2: Molecule Tokenizer Pretraining.**  We connect the Causal Q-Former with subsequent blocks and use the objective defined in Equation (2). We employ the pretrained ChemFormer [17] as the generative model. Specifically, we leverage the SMILES encoder and SMILES decoder provided by ChemFormer. The molecule codebook size is set to $K = 2048$. As shown in Figure 2, we keep the molecule encoder, SMILES encoder, and SMILES decoder frozen, while updating the Causal Q-Former, codebook, and adapter.

**Stage-3: Unified Molecule-Text Pretraining.**  We integrate the molecule tokenizer with the LLM using the unified vocabulary of molecule tokens and text tokens. We employ the LM objective defined in Equation (3) to pretrain the LLM. Pretraining involves molecule-to-text autoregression and text-to-molecule autoregression, aimed at enhancing UniMoT's multi-modal comprehension and generation capabilities. To enhance efficiency, we train the LLM using LoRA tuning [14].

**Stage-4: Task-Specific Instruction Tuning.**  UniMoT is fine-tuned on seven comprehension and generation tasks: molecular property prediction, molecule captioning, molecule-text retrieval, caption-guided molecule generation, reagent prediction, forward reaction prediction, and retrosynthesis. We also utilize LoRA tuning to improve efficiency. This stage ensures UniMoT can accurately interpret and respond to human instructions, making it versatile and effective for molecular tasks.

## 4   Experiments

### 4.1   Molecule Comprehension Tasks

**Molecular Property Prediction Task.**  The goal of molecular property prediction is to forecast a molecule's intrinsic physical and chemical properties. For the classification task, we incorporate eight binary classification datasets from MoleculeNet [53]. Models are tasked with generating a single prediction ("yes" or "no"). We compare UniMoT with the following baselines: KV-PLM [58], AttrMask [16], InfoGraph [40], MolCLR [48], GraphMVP [30], MoleculeSTM [31], and InstructMol [6]. The ROC-AUC (%) results on the MoleculeNet datasets are shown in Table 1. The performance of the regression task of molecular property prediction is provided in Appendix D. Compared to traditional graph learning methods and molecular LLMs like InstructMol, UniMoT demonstrates consistent improvements across the eight datasets, indicating its robust molecule comprehension abilities.

**Molecule Captioning Task.**  The molecule captioning task involves generating a comprehensive description of a molecule. We compare UniMoT with several baselines: MolT5 [11], MoMu [39], InstructMol [6], MolCA [33], and 3D-MoLM [25]. BLEU [37], ROUGE [27], and METEOR [1] are adopted as evaluation metrics. UniMoT is evaluated for molecule captioning on the PubChem and

Table 1: ROC-AUC (%) of molecular property prediction task (classification) on the MoleculeNet datasets. **Bold** indicates the best performance and underline indicates the second best performance.

| Model | BBBP↑ | Tox21↑ | ToxCast↑ | Sider↑ | ClinTox↑ | MUV↑ | HIV↑ | BACE↑ |
|---|---|---|---|---|---|---|---|---|
| KV-PLM [58] | 70.50 | 72.12 | 55.03 | 59.83 | 89.17 | 54.63 | 65.40 | 78.50 |
| AttrMask [16] | 67.79 | 75.00 | 63.57 | 58.05 | 75.44 | 73.76 | 75.44 | 80.28 |
| InfoGraph [40] | 64.84 | 76.24 | 62.68 | 59.15 | 76.51 | 72.97 | 70.20 | 77.64 |
| MolCLR [48] | 67.79 | 75.55 | 64.58 | 58.66 | 84.22 | 72.76 | 75.88 | 71.14 |
| GraphMVP [30] | 68.11 | **77.06** | 65.11 | 60.64 | 84.46 | 74.38 | 77.74 | 80.48 |
| MoleculeSTM [31] | 69.98 | 76.91 | 65.05 | **60.96** | 92.53 | 73.40 | 76.93 | 80.77 |
| InstructMol (Vicuna-7B) [6] | 70.00 | 74.67 | 64.29 | 57.80 | 91.48 | 74.62 | 68.90 | 82.30 |
| UniMoT (LLaMA2-7B) | **71.37** | 76.43 | **65.78** | 59.79 | **92.89** | **75.97** | **78.49** | **83.69** |

Table 2: Performance (%) of molecule captioning task on the PubChem dataset. **Bold** indicates the best performance and underline indicates the second best performance.

| Model | BLEU-2↑ | BLEU-4↑ | ROUGE-1↑ | ROUGE-2↑ | ROUGE-L↑ | METEOR↑ |
|---|---|---|---|---|---|---|
| MolT5-Small (T5-Small) [11] | 22.5 | 15.2 | 30.4 | 13.5 | 20.3 | 24.0 |
| MolT5-Base (T5-Base) [11] | 24.5 | 16.6 | 32.2 | 14.0 | 21.4 | 26.1 |
| MolT5-Large (T5-Large) [11] | 25.9 | 17.3 | 34.1 | 16.4 | 23.4 | 28.0 |
| MoMu-Small (T5-Small) [39] | 22.9 | 16.0 | 31.0 | 13.7 | 20.8 | 24.4 |
| MoMu-Base (T5-Base) [39] | 24.7 | 16.8 | 32.5 | 14.6 | 22.1 | 27.2 |
| MoMu-Large (T5-Large) [39] | 26.3 | 18.0 | 34.8 | 16.9 | 24.8 | 28.7 |
| InstructMol (Vicuna-7B) [6] | 18.9 | 11.7 | 27.3 | 11.8 | 17.8 | 21.3 |
| MolCA (OPT-125M) [33] | 25.9 | 17.5 | 34.4 | 16.6 | 23.9 | 28.5 |
| MolCA (OPT-1.3B) [33] | 28.6 | 21.3 | 36.2 | 21.4 | 29.7 | 32.6 |
| 3D-MoLM (LLaMA2-7B) [25] | 30.3 | 22.5 | 36.8 | 22.3 | 31.2 | 33.1 |
| UniMoT (LLaMA2-7B) | **31.3** | **23.8** | **37.5** | **23.7** | **33.6** | **34.8** |

CheBI-20 datasets. Performance on the PubChem dataset is shown in Table 2, while the performance on the CheBI-20 dataset and some concrete examples are presented in Appendix D.

From Table 2, we observe that UniMoT consistently outperforms the baselines by a significant margin. This task is more complex than classification or regression, providing a robust measure of the model's molecule comprehension abilities. Notably, our proposed tokenizer-based architecture surpasses the projection-based architecture (such as InstructMol), Q-Former-based architecture (such as MolCA and 3D-MoLM), and models trained with contrastive learning strategies (such as MoMu). The results demonstrate that the molecule tokenizer can generate molecule tokens with high-level molecular and textual information, enhancing molecule comprehension abilities.

**Molecule-Text Retrieval Task.** The molecule-text retrieval task involves using a molecule to retrieve text (M2T) and using text to retrieve a molecule (T2M). We compare UniMoT with several baselines: Sci-BERT [2], KV-PLM [58], MoMu [39], MoleculeSTM [31], MolCA [33], and 3D-MoLM [25]. We report the performance of retrieval using a batch of 64 random samples and the entire test set, evaluated with the metrics of Accuracy and Recall@20. We use the checkpoint from Stage-1 of pretraining. UniMoT is evaluated on the datasets of PubChem, PCdes, and MoMu. Performance on the PubChem dataset is shown in Table 3, while performance on the PCdes and MoMu datasets is presented in Appendix D. UniMoT can understand complex molecule-text interactions through the introduction of the Causal Q-Former. From Table 3, UniMoT demonstrates superior performance over the baselines on molecule-text retrieval, particularly in molecule-to-text retrieval. This underscores UniMoT's capability in learning fine-grained alignment between molecules and text.

### 4.2 Molecule Generation Tasks

We employ molecule generation tasks, which encompass caption-guided molecule generation, reagent prediction, forward reaction prediction, and retrosynthesis. Caption-guided molecule generation involves generating molecular structures based on textual descriptions. Reagent prediction entails determining suitable reagents given reactants and products. Forward reaction prediction involves predicting probable products given specific reactants and reagents. Retrosynthesis involves deconstructing a target molecule into simpler starting materials. We compare UniMoT with the following

Table 3: Performance (%) of molecule-text retrieval task on the PubChem dataset. **Bold** indicates the best performance and underline indicates the second best performance.

| Model | Retrieval in batch | | | | Retrieval in test set | | | |
| | M2T (%) | | T2M (%) | | M2T (%) | | T2M (%) | |
| | Acc↑ | R@20↑ | Acc↑ | R@20↑ | Acc↑ | R@20↑ | Acc↑ | R@20↑ |
|---|---|---|---|---|---|---|---|---|
| Sci-BERT [2] | 85.3 | 98.7 | 84.2 | 98.4 | 41.7 | 87.3 | 40.2 | 86.8 |
| KV-PLM [58] | 86.1 | 98.6 | 85.2 | 98.5 | 42.8 | 88.5 | 41.7 | 87.8 |
| MoMu (Sci-BERT) [39] | 87.6 | 99.2 | 86.4 | 99.4 | 47.3 | 90.8 | 48.1 | 89.9 |
| MoMu (KV-PLM) [39] | 88.2 | 99.4 | 87.3 | 99.4 | 48.5 | 91.6 | 49.5 | 90.7 |
| MoleculeSTM [31] | 90.5 | 99.6 | 88.6 | 99.5 | 52.7 | 92.9 | 53.2 | 92.5 |
| MolCA (OPT-1.3B) [33] | 92.6 | 99.8 | 91.3 | 99.5 | 67.9 | 94.4 | 68.6 | 93.3 |
| 3D-MoLM (LLaMA2-7B) [25] | 93.5 | **100.0** | **92.9** | **99.6** | 69.1 | 95.9 | **70.1** | **94.9** |
| UniMoT (LLaMA2-7B) | **93.6** | **100.0** | 92.7 | 99.4 | **69.5** | **96.3** | 69.8 | 94.4 |

Table 4: Performance of molecule generation tasks on the Mol-Instructions dataset, including caption-guided molecule generation, reagent prediction, forward reaction prediction, and retrosynthesis. **Bold** indicates the best performance, and underline indicates the second best performance.

| Model | Exact↑ | BLEU↑ | Levenshtein↓ | RDK FTS↑ | MACCS FTS↑ | Morgan FTS↑ | Validity↑ |
|---|---|---|---|---|---|---|---|
| **Caption-guided Molecule Generation** | | | | | | | |
| LLaMA [44] | 0.000 | 0.003 | 59.864 | 0.005 | 0.000 | 0.000 | 0.003 |
| Vicuna [7] | 0.000 | 0.006 | 60.356 | 0.006 | 0.001 | 0.000 | 0.001 |
| Mol-Instructions [12] | 0.002 | 0.345 | 41.367 | 0.231 | 0.412 | 0.147 | 1.000 |
| MolT5 [11] | 0.112 | 0.546 | 38.276 | 0.400 | 0.538 | 0.295 | 0.773 |
| UniMoT | **0.237** | **0.698** | **27.782** | **0.543** | **0.651** | **0.411** | 1.000 |
| **Reagent Prediction** | | | | | | | |
| LLaMA [44] | 0.000 | 0.003 | 28.040 | 0.037 | 0.001 | 0.001 | 0.001 |
| Vicuna [7] | 0.000 | 0.010 | 27.948 | 0.038 | 0.002 | 0.001 | 0.007 |
| Mol-Instructions [12] | 0.044 | 0.224 | 23.167 | 0.237 | 0.364 | 0.213 | 1.000 |
| InstructMol [6] | 0.129 | 0.610 | 19.664 | 0.444 | 0.539 | 0.400 | 1.000 |
| UniMoT | **0.167** | **0.728** | **14.588** | **0.549** | **0.621** | **0.507** | 1.000 |
| **Forward Reaction Prediction** | | | | | | | |
| LLaMA [44] | 0.000 | 0.020 | 42.002 | 0.001 | 0.002 | 0.001 | 0.039 |
| Vicuna [7] | 0.000 | 0.057 | 41.690 | 0.007 | 0.016 | 0.006 | 0.059 |
| Mol-Instructions [12] | 0.045 | 0.654 | 27.262 | 0.313 | 0.509 | 0.262 | 1.000 |
| InstructMol [6] | 0.536 | 0.967 | 10.851 | 0.776 | 0.878 | 0.741 | 1.000 |
| UniMoT | **0.611** | **0.980** | **8.297** | **0.836** | **0.911** | **0.807** | 1.000 |
| **Retrosynthesis** | | | | | | | |
| LLaMA [44] | 0.000 | 0.036 | 46.844 | 0.018 | 0.029 | 0.017 | 0.010 |
| Vicuna [7] | 0.000 | 0.057 | 46.877 | 0.025 | 0.030 | 0.021 | 0.017 |
| Mol-Instructions [12] | 0.009 | 0.705 | 31.227 | 0.283 | 0.487 | 0.230 | 1.000 |
| InstructMol [6] | 0.407 | 0.941 | 13.967 | 0.753 | 0.852 | 0.714 | 1.000 |
| UniMoT | **0.478** | **0.974** | **11.634** | **0.810** | **0.909** | **0.771** | 1.000 |

baselines: LLaMA [44], Vicuna [7], Mol-Instructions [12], and InstructMol [6]. The metrics used to evaluate molecule generation tasks include Exact Match, BLEU [37], Levenshtein Distance [22], RDKit Fingerprint Similarity [20], MACCS Fingerprint Similarity [10], and Morgan Fingerprint Similarity [36]. These metrics evaluate structural similarity between generated and target molecules, along with Validity [19], which assesses the proportion of chemically valid molecules generated. We utilize the Mol-Instructions dataset to evaluate the generation capabilities of UniMoT, and the results are presented in Table 4.

As the baselines generate SMILES strings and then convert them to molecules, UniMoT directly leverages the generated molecule tokens and obtains their embeddings from the learned codebook. These embeddings can be decoded to desired molecules through the pretrained adapter and SMILES decoder. Regarding the results in Table 4, UniMoT exhibits the capability to generate valid molecules with a higher degree of similarity to the target molecules compared to the baselines. UniMoT can generate molecules as if they were text, demonstrating strong generation capabilities and providing a new perspective to molecule generation tasks.

Table 5: Ablation study on the projector and representation form for the molecule captioning task using the PubChem dataset.

| Projector | Input to LLM | BLEU-2 | BLEU-4 | ROUGE-1 | ROUGE-2 | ROUGE-L | METEOR |
|---|---|---|---|---|---|---|---|
| Projection Layer | Molecule Emb. | 19.3 | 12.1 | 27.9 | 12.3 | 18.1 | 21.5 |
| Q-Former | Query Emb. | 28.6 | 21.3 | 36.2 | 21.4 | 29.7 | 32.6 |
| Causal Q-Former | Causal Emb. | 32.8 | 25.2 | 39.2 | 24.8 | 35.3 | 36.5 |
| Causal Q-Former | Causal Tokens | 31.3 | 23.8 | 37.5 | 23.7 | 33.6 | 34.8 |

Table 6: Ablation study on the model size and tuning strategy for the molecule captioning task using the PubChem dataset.

| Model Size | Tuning | BLEU-2 | BLEU-4 | ROUGE-1 | ROUGE-2 | ROUGE-L | METEOR |
|---|---|---|---|---|---|---|---|
| LLaMA2-7B | LoRA Tuning | 31.3 | 23.8 | 37.5 | 23.7 | 33.6 | 34.8 |
| LLaMA2-7B | Fully Tuning | 32.0 | 24.6 | 38.3 | 24.3 | 34.7 | 35.6 |
| LLaMA2-13B | LoRA Tuning | 31.8 | 24.3 | 38.0 | 24.1 | 34.4 | 35.3 |

## 4.3 Ablation Studies

**Cross-Modal Projector.** We conducted an ablation study on the cross-modal projector, with the results on the molecule captioning task shown in Table 5. The linear projection demonstrated the worst performance, indicating that the molecule features lack textual information, thus hindering effective molecule-text interactions and alignment. Additionally, we compared the performance of a Q-Former with bidirectional self-attention to a Causal Q-Former with causal self-attention. The results show that query embeddings with causal dependency outperform those with bidirectional dependency. This demonstrates that input with left-to-right causal dependency aligns with the unidirectional attention mechanism in LLMs, leading to improved performance.

**Discrete vs. Continuous Representation.** We compare the performance of continuous causal query embeddings and discrete tokens quantized from causal embeddings as inputs to LLMs. As shown in Table 5, continuous embeddings demonstrate better performance than discrete tokens in understanding molecules. This result is reasonable since the quantization process causes information loss in discrete tokens. However, we still use discrete token representation to facilitate the autoregressive training paradigm of LLMs, which supports the unification of comprehension and generation tasks. To achieve this unification, we unavoidably sacrifice some performance in comprehension tasks.

**Model Size and Tuning Stategy.** We conducted a comparison of molecule captioning performance across various model sizes and tuning strategies, as illustrated in Table 6. Our findings indicate that scaling up the LLM to 13B or adopting a fully tuning strategy yields only marginal improvements in performance compared to using LLaMA2-7B with LoRA tuning. While larger models and fully tuning strategies might offer slight gains in performance, they come at a significant cost in terms of efficiency. Considering the trade-off between achieving high performance and maintaining efficiency, we have chosen to utilize LLaMA2-7B with LoRA tuning in our experiments. This ensures that our model remains both powerful and practical.

## 5 Conclusion

This work introduces UniMoT, an innovation in the field of molecular-textual understanding and generation, which has successfully unified these two distinct modalities under a single, coherent framework. By integrating a Vector Quantization-driven tokenizer with a Causal Q-Former, UniMoT overcomes previous architectural limitations where molecule and text modalities were not treated equally, lacking a dedicated supervision signal for the molecular domain. This unique tokenizer transforms molecules into sequences of discrete tokens, embedding high-level molecular and textual information cohesively. Moreover, by employing a four-stage training scheme, UniMoT has emerged as a versatile multi-modal LLM, adept at handling molecule-to-text and text-to-molecule tasks. Extensive empirical evaluations demonstrate that UniMoT attains state-of-the-art performance across a diverse array of molecule comprehension and generation tasks.

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

## A   Details of Causal Q-Former

The Q-Former operates as a query-based transformer that utilizes learnable query vectors to interact with molecule features extracted by a frozen encoder. These queries are essential for extracting relevant information from the molecule features. The Q-Former comprises both a molecule transformer and a text transformer, sharing self-attention layers. The text transformer architecture is based on BERT [9], while the molecule transformer incorporates cross-attention layers between self-attention and feed-forward layers. Q-Former employs a cross-attention mechanism where the query vectors selectively attend to different aspects of the molecule features, allowing the model to capture critical details necessary for understanding and generating textual descriptions of molecular properties.

Specifically, we incorporate causal masks into the queries, ensuring that they only interact with preceding queries. This ensures the sequence of query embeddings maintains a causal dependency, aligning with the requirements of LLMs operating on text sequence. The Causal Q-Former is illustrated in Figure 4. We employ the Causal Q-Former to generate causal query embeddings $\mathbf{Z} = \{z_i\}_{i=1}^M \in \mathbb{R}^{M \times d}$ containing high-level molecular and textual information, where $M$ represents the number of queries and $d$ denotes the dimension of query embeddings. Next, we introduce three tailored objectives MTC, MTM, and MTG for the pretraining of the Causal Q-Former.

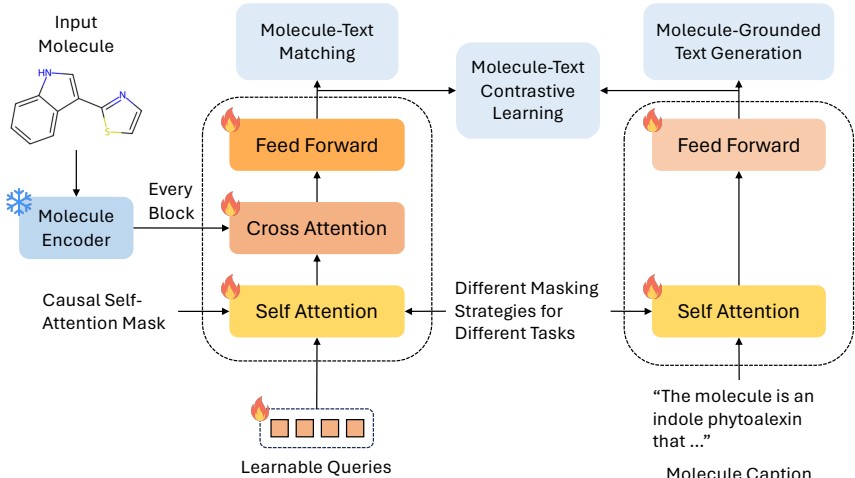

Figure 4: Illustration of our proposed Causal Q-Former. The Causal Q-Former provides causal query embeddings for subsequent blocks.

**Molecule-Text Contrastive Learning (MTC)** aims to align molecule and text features by maximizing their mutual information. This is achieved by maximizing the molecule-text similarity of positive pairs against that of negative pairs. We utilize the last query embedding $z_M$ of the query sequence $\{z_i\}_{i=1}^M$ as the query representation, since the output query sequence is causal and the last embedding contains global information from the queries. For text representation, we use the output embedding of the [CLS] token, denoted as $\mathbf{y}$. The contrastive learning loss is expressed as follows:

$$\mathcal{L}_{\text{MTC}} = -\frac{1}{B}\sum_{i=1}^{B}\log\frac{\exp((\mathbf{z}_M^i)^T\mathbf{y}^i/\tau)}{\sum_{j=1}^{B}\exp((\mathbf{z}_M^i)^T\mathbf{y}^j/\tau)} - \frac{1}{B}\sum_{i=1}^{B}\log\frac{\exp((\mathbf{y}^i)^T\mathbf{z}_M^i/\tau)}{\sum_{j=1}^{B}\exp((\mathbf{y}^i)^T\mathbf{z}_M^j/\tau)}, \quad (4)$$

where $B$ denotes the batch size, and $\tau$ represents the temperature parameter. Here, $\mathbf{z}_M^i$ and $\mathbf{y}^i$ refer to the $i$-th query and text representations in a batch, respectively.

**Molecule-Text Matching (MTM)** focuses on learning fine-grained alignment between molecule and text features. As query embeddings $\mathbf{Z} = \{z_i\}_{i=1}^M$ capture both molecular and textual information through cross-attention and self-attention layers respectively, we utilize the last query embedding $z_M$ as input to a binary classifier. This classifier predicts whether a given molecule-text pair is matched or unmatched. The corresponding loss function is formulated as follows:

$$\mathcal{L}_{\text{MTM}} = -\frac{1}{B}\sum_{i=1}^{B}\log\frac{\exp(\phi(\mathbf{z}_M \mid \mathbf{X}^i, \mathbf{t}^i))}{\sum_{j=1}^{B}\exp(\phi(\mathbf{z}_M \mid \mathbf{X}^i, \mathbf{t}^j)) + \sum_{j=1}^{B}\exp(\phi(\mathbf{z}_M \mid \mathbf{X}^j, \mathbf{t}^i))}, \quad (5)$$

where $\phi$ represents a binary classifier, and $\mathbf{X}^i$ and $\boldsymbol{t}^i$ denote the $i$-th input molecule features and input text in a batch, respectively.

**Molecule-grounded Text Generation (MTG)** focuses on generating textual descriptions given a molecule input. In this task, causal masks for queries are not applied since only textual output is required. However, causal masks are applied for text, allowing each text token to attend to its preceding text tokens and all queries, but not subsequent tokens. The Language Modeling (LM) loss function is applied to model the generation of text $\boldsymbol{t}^i$ conditioned on the molecule input $\mathbf{X}^i$, formulated as:

$$\mathcal{L}_{\text{MTG}} = -\frac{1}{B} \sum_{i=1}^{B} \sum_{j=1}^{L} \log p\left(t_j^i \mid t_1^i, \cdots, t_{j-1}^i, \mathbf{X}^i\right), \tag{6}$$

where $t_j^i$ represents the $j$-th token in the text sequence $\boldsymbol{t}^i$. Here, $\mathbf{X}^i$ and $\boldsymbol{t}^i$ denote the $i$-th input molecule features and generated text in a batch, respectively.

The total loss for training the Q-Former encompasses the three aforementioned objectives:

$$\mathcal{L}_{\text{Q-Former}} = \mathcal{L}_{\text{MTC}} + \mathcal{L}_{\text{MTM}} + \mathcal{L}_{\text{MTG}}. \tag{7}$$

# B  Details of Datasets

This section provides detailed information about the datasets used in evaluating the performance of UniMoT across various tasks. The datasets are utilized for molecular property prediction, molecule captioning, molecule-text retrieval, and molecule generation tasks. Each dataset serves a unique purpose in assessing different capabilities of the model.

We present the details of the Molecular Property Prediction Datasets below:

- **BBBP** [53]: The Blood-Brain Barrier Penetration dataset predicts the ability of molecules to penetrate the blood-brain barrier.
- **Tox21** [53]: This dataset is part of the Toxicology in the 21st Century initiative, used for toxicity prediction.
- **ToxCast** [53]: Another toxicity prediction dataset with a broader range of biological assays.
- **Sider** [53]: Side Effect Resource database, used for predicting drug side effects.
- **ClinTox** [53]: Clinical Toxicity dataset for predicting clinical trial toxicity outcomes.
- **MUV** [53]: Maximum Unbiased Validation dataset for virtual screening.
- **HIV** [53]: Human Immunodeficiency Virus dataset for predicting anti-HIV activities.
- **BACE** [53]: Beta-Secretase 1 dataset for predicting inhibitors of the BACE-1 enzyme, relevant for Alzheimer's research.
- **QM9** [12]: The quantum mechanics properties dataset, where the objective is to predict key quantum mechanics properties of a given molecule, such as HUMO, LUMO, and the HUMO-LUMO gap.

We present the details of the Molecule Captioning Datasets below:

- **PubChem** [18]: A large dataset of chemical molecules used for generating textual descriptions of molecular structures.
- **ChEBI-20** [11]: A subset of the Chemical Entities of Biological Interest database, provides structured and detailed descriptions of molecules, enhancing the model's ability to generate accurate captions.

We present the details of the Molecule-Text Retrieval Datasets below:

- **PubChem** [18]: Used for both molecule-to-text (M2T) and text-to-molecule (T2M) retrieval tasks.
- **PCdes** [58]: Another dataset for evaluating M2T and T2M retrieval accuracy.
- **MoMu** [39]: Dataset specifically designed for molecule-text interactions and retrieval tasks.

Table 7: Summary of datasets, their types, tasks, descriptions, URLs, and licenses used for evaluating UniMoT.

| Dataset | Type | Tasks | Description | URL | License |
|---|---|---|---|---|---|
| BBBP | Classification | Molecular Property Prediction | Predicts blood-brain barrier penetration ability. | BBBP URL | CC-BY 4.0 |
| Tox21 | Classification | Molecular Property Prediction | Toxicity prediction using the Tox21 initiative data. | Tox21 URL | Public Domain |
| ToxCast | Classification | Molecular Property Prediction | Broad toxicity prediction with various biological assays. | ToxCast URL | Public Domain |
| Sider | Classification | Molecular Property Prediction | Predicts drug side effects. | Sider URL | CC-BY 4.0 |
| ClinTox | Classification | Molecular Property Prediction | Clinical trial toxicity prediction. | ClinTox URL | Public Domain |
| MUV | Classification | Molecular Property Prediction | Virtual screening for unbiased validation. | MUV URL | CC-BY 4.0 |
| HIV | Classification | Molecular Property Prediction | Predicts anti-HIV activity of molecules. | HIV URL | Public Domain |
| BACE | Classification | Molecular Property Prediction | Predicts inhibitors of the BACE-1 enzyme. | BACE URL | Public Domain |
| QM9 | Regression | Molecular Property Prediction | Predicts various molecular properties such as atomization energy, dipole moment, etc. | QM9 URL | CC-BY 4.0 |
| PubChem | Captioning, Retrieval | Molecule Captioning, Molecule-Text Retrieval | Generates descriptions and retrieves text/molecules based on input molecules/text. | PubChem URL | Public Domain |
| ChEBI-20 | Captioning | Molecule Captioning | Generates detailed descriptions of molecular structures. | ChEBI-20 URL | CC-BY 4.0 |
| PCdes | Retrieval | Molecule-Text Retrieval | Used for evaluating accuracy in molecule-text retrieval tasks. | PCdes URL | CC-BY 4.0 |
| MoMu | Retrieval | Molecule-Text Retrieval | Dataset for molecule-text interaction and retrieval evaluation. | MoMu URL | CC-BY 4.0 |
| Mol-Instructions | Generation | Molecule Generation | Includes tasks such as molecule generation from descriptions, reagent prediction, etc. | Mol-Instructions URL | CC-BY 4.0 |

We present the details of the Molecule Generation Datasets below:

- **Mol-Instructions** [12]: This dataset includes tasks such as caption-guided molecule generation, reagent prediction, forward reaction prediction, and retrosynthesis. It is used to evaluate the model's ability to generate molecular structures based on textual descriptions and other related tasks.

We summarize the datasets used for evaluating UniMoT in Table 7. It encompasses various types of datasets, including those for classification, regression, captioning, retrieval, and generation tasks. Each dataset is described in terms of its type, tasks it supports, a brief description of its content, its URL for access, and the license under which it is distributed. The licenses vary, with some datasets being in the public domain and others under CC-BY 4.0 license.

## C   Details of Training

**Stage-1: Causal Q-Former Pretraining.**   During Stage-1, we only connect the molecule encoder and the Causal Q-Former, leaving out other blocks. We leverage the pretrained molecule encoder from MoleculeSTM [31], which has undergone extensive contrastive learning with molecule-text pairs. We utilize the PubChem dataset [18] for pretraining, keeping the molecule encoder frozen while updating only the Causal Q-Former. Both queries and text serve as input to the Causal Q-Former, while only queries serve as input in subsequent stages. Inspired by BLIP-2 [23], we employ three tailored objectives – Molecule-Text Contrastive Learning (MTC), Molecule-Text Matching (MTM), and Molecule-grounded Text Generation (MTG) – for the pretraining of the Causal Q-Former, as detailed in Appendix A.

The dimension of molecule features is set to 300. We use 16 queries, each with a dimension of 768. The size of $\mathbf{Z}$ ($16 \times 768$) is much smaller than the size of molecule features $\mathbf{X}$ (e.g., $150 \times 300$). The Q-former is pretrained for 50 epochs. We adopt the AdamW optimizer with a weight decay of 0.05, and a cosine decay learning rate scheduler, with a minimal learning rate of 1e-5. The batch size is set to 64. The computational overhead for this pretraining is 20 GPU hours on 4 NVIDIA A100 GPUs.

**Stage-2: Molecule Tokenizer Pretraining.**   We connect the Causal Q-Former with the subsequent blocks and train the molecule tokenizer using the objective defined in Equation (2). Following the approach of RetMol [49], we utilize SMILES strings [50] to represent molecules, and employ the pretrained ChemFormer [17] as the generative model. Specifically, we leverage the SMILES encoder and SMILES decoder components provided by ChemFormer. We utilize PubChem [18] and CheBI-20 [11] datasets, keeping the molecule encoder, SMILES encoder, and SMILES decoder frozen, while updating the Causal Q-Former, codebook, and adapter. Once optimized, the molecule tokenizer remains unchanged throughout the subsequent stages.

The molecule codebook size is set to $K = 2048$, and the dimension of codebook embedding is 768. The tokenizer is pretrained for 50 epochs. We adopt the AdamW optimizer with a weight decay of 0.05, and a cosine decay learning rate scheduler, with a minimal learning rate of 1e-5. The batch size is set to 64. The computational overhead for this pretraining is 40 GPU hours on 4 NVIDIA A100 GPUs.

**Stage-3: Unified Molecule-Text Pretraining.**   We connect the molecule tokenizer with the LLM and employ the LM objective defined in Equation (3) to pretrain the LLM. We utilize LLaMA [44] as the default LLM. To construct the unified molecule-text vocabulary, we merge 2048 molecule codes with the original text vocabulary. Pretraining the LLM involves molecule-to-text autoregression and text-to-molecule autoregression, aimed at enhancing UniMoT's multi-modal comprehension and generation capabilities. We utilize datasets PubChem [18], CheBI-20 [11], PCdes [58], and MoMu [39] for this purpose. To enhance efficiency, we train the LLM using LoRA tuning [14].

The multi-modal LLM is pretrained for 10 epochs. We adopt the AdamW optimizer with a weight decay of 0.05, and a cosine decay learning rate scheduler, with a minimal learning rate of 1e-5. The batch size is set to 32. The computational overhead for this pretraining is 50 GPU hours on 4 NVIDIA A100 GPUs. To reduce CUDA memory usage, we integrate LoRA with the parameters set to $r = 8$, $\alpha = 32$, and dropout = 0.1. This integration is applied to the `k_proj`, `v_proj`, `q_proj`, and `o_proj` modules.

Table 8: Instruction samples for comprehension and generation tasks: molecular property prediction, molecule captioning, molecule-text retrieval, caption-guided molecule generation, reagent prediction, forward reaction prediction, and retrosynthesis.

| Task | Instruction |
|---|---|
| Molecular Property Prediction (Regression) | Instruction: *Could you give me the LUMO energy value of this molecule?* (Optional: The SMILES sequence is: SMILES) Output: *0.0576.* |
| Molecular Property Prediction (Classification) | Instruction: *Evaluate whether the given molecule is able to enter the blood-brain barrier.* (Optional: The SMILES sequence is: SMILES) Output: *Yes.* |
| Molecule Captioning | Instruction: *Could you give me a brief overview of this molecule?* (Optional: The SMILES sequence is: SMILES) Output: *The molecule is an indole phytoalexin that ...* |
| Molecule-Text Retrieval | Instruction: *Retrieve relevant text for the given molecule.* (Optional: The SMILES sequence is: SMILES) Output: *The molecule is associated with ...* |
| Caption-Guided Molecule Generation | Instruction: *Create a molecule with the structure as described: The molecule is a primary arylamine that ...* Output: SMILES *of the molecule.* |
| Reagent Prediction | Instruction: *Please provide possible reagents based on the following chemical reaction.* <REACTANT A> <REACTANT B> ... ⤚ <PRODUCTs> Output: SMILES *of the reagents.* |
| Forward Reaction Prediction | Instruction: *With the provided reactants and reagents, propose a potential product:* <REACTANT A> <REACTANT B> ... <REAGENT A> <REAGENT B> ... Output: SMILES *of the products.* |
| Retrosynthesis | Instruction: *Please suggest potential reactants used in the synthesis of the product:* <PRODUCTs> Output: SMILES *of the reactants and reagents.* |

**Stage-4: Task-Specific Instruction Tuning.** We perform instruction tuning to align UniMoT with human instructions through supervised fine-tuning on seven tasks: molecular property prediction, molecule captioning, molecule-text retrieval, caption-guided molecule generation, reagent prediction, forward reaction prediction, and retrosynthesis. For the molecular property prediction task, we utilize the quantum mechanics properties dataset [12] for regression prediction and the MoleculeNet datasets [53] for property classification. For the molecule captioning and molecule-text retrieval tasks, we employ datasets PubChem [18], CheBI-20 [11], PCdes [58], and MoMu [39]. For the remaining tasks, we utilize the Mol-Instructions dataset [12] to conduct instruction tuning. We fine-tune UniMoT for 10 epochs on each task using the same optimizer, learning rate scheduler, and LoRA configurations as in Stage-3 pretraining. Instruction samples for comprehension and generation tasks are shown in Table 8.

We have summarized the detailed training hyperparameters of UniMoT in Table 9.

# D   Details and More Results of Experiments

**Molecular Property Prediction Task.** Property prediction aims to anticipate a molecule's intrinsic physical and chemical properties based on its structural or sequential characteristics. In the regression task, we conduct experiments on the quantum mechanics properties dataset QM9 [12], where the objective is to predict key quantum mechanics properties of a given molecule, such as HUMO, LUMO, and the HUMO-LUMO gap. We compare UniMoT against several baselines, including Alpaca [42], Baize [55], LLaMA2-7B [44], Vicuna-13B [7], Mol-Instructions [12], and InstructMol [6]. Mean Absolute Error (MAE) serves as our evaluation metric. The performance of the regression task on the QM9 dataset is presented in Table 10. Compared to previous single-modal instruction-tuned LLMs and molecular LLMs, UniMoT exhibits further improvement on the regression task, showcasing its fundamental comprehension abilities in molecular contexts.

**Molecule Captioning Task.** The molecule captioning task involves generating a comprehensive description of a molecule. For this task, we compare UniMoT with several baselines: MolT5 [11],

Table 9: The detailed training hyperparameters of UniMoT.

| Configuration | Q-Former Pretraining | Tokenizer Pretraining | LLM Pretraining |
|---|---|---|---|
| Molecule Encoder | MoleculeSTM | MoleculeSTM | MoleculeSTM |
| SMILES Encoder | - | ChemFormer | ChemFormer |
| SMILES Decoder | - | ChemFormer | ChemFormer |
| LLM Base | - | - | LLaMA2-7B |
| Epoch | 50 | 50 | 10 |
| Optimizer | AdamW | AdamW | AdamW |
| Codebook Size | 2048 | 2048 | 2048 |
| Number of Queries | 16 | 16 | 16 |
| Query Embedding Dim. | 768 | 768 | 768 |
| Molecule Embedding Dim. | 300 | 300 | 300 |
| Batch Size | 64 | 64 | 32 |
| Minimal Learning Rate | 1e-5 | 1e-5 | 1e-5 |
| Learning Rate Scheduler | Cosine | Cosine | Cosine |
| Warm-up Steps | 1000 | 1000 | 1000 |
| Weight Decay | 0.05 | 0.05 | 0.05 |
| LoRA Config | - | - | $r = 8, \alpha = 32, \text{dropout} = 0.1$ |
| Precision | bfloat16 | bfloat16 | bfloat16 |
| GPU Usage | 4 NVIDIA A100 | 4 NVIDIA A100 | 4 NVIDIA A100 |
| Training Time | 20 GPU hours | 40 GPU hours | 50 GPU hours |

Table 10: Mean Absolute Error (MAE) of molecular property prediction task (regression) on the QM9 dataset. **Bold** indicates the best performance and underline indicates the second best performance. $\Delta\epsilon$ is the HOMO-LUMO energy gap.

| Model | HOMO↓ | LUMO↓ | $\Delta\epsilon$ ↓ | AVG↓ |
|---|---|---|---|---|
| Alpaca (LLaMA-7B) [42] | - | - | - | 322.109 |
| Baize (LLaMA-7B) [55] | - | - | - | 261.343 |
| LLaMA2-7B [44] | 0.7367 | 0.8641 | 0.5152 | 0.7510 |
| Vicuna-13B [7] | 0.7135 | 3.6807 | 1.5407 | 1.9783 |
| Mol-Instructions (LLaMA-7B) [12] | 0.0210 | 0.0210 | 0.0203 | 0.0210 |
| InstructMol (Vicuna-7B) [6] | 0.0048 | 0.0050 | 0.0061 | 0.0050 |
| UniMoT (LLaMA2-7B) | **0.0042** | **0.0047** | **0.0055** | **0.0049** |

MoMu [39], InstructMol [6], MolCA [33], and 3D-MoLM [25]. We adopt BLEU [37], ROUGE [27], and METEOR [1] as the evaluation metrics. The performance of UniMoT in the molecule captioning task on the CheBI-20 dataset is presented in Table 11. Some concrete examples of molecule captioning task are presented in Table 12. From the results, it is evident that UniMoT consistently outperforms the baselines by a significant margin. These results underscore the effectiveness of the molecule tokenizer in providing molecule tokens with high-level molecular and textual information, thus enhancing molecule comprehension.

**Molecule-Text Retrieval Task.** The molecule-text retrieval task involves using a molecule to retrieve text (M2T) and using text to retrieve a molecule (T2M). We compare UniMoT with several baselines: Sci-BERT [2], KV-PLM [58], MoMu [39], MoleculeSTM [31], MolCA [33], and 3D-MoLM [25]. We report the performance of retrieval using a batch of 64 random samples and the entire test set, evaluated with the metrics of Accuracy and Recall@20. We use the checkpoint from Stage-1 of pretraining. Performance on the PCdes and MoMu datasets is shown in Table 13. UniMoT demonstrates superior performance over the baselines on molecule-text retrieval, particularly in molecule-to-text retrieval. This demonstrates that UniMoT has learned fine-grained alignment between molecules and text, and it can understand molecule-text interactions through the introduction of the Causal Q-Former.

**Molecule Generation Tasks.** Molecule generation tasks include caption-guided molecule generation, reagent prediction, forward reaction prediction, and retrosynthesis.

Table 11: Performance (%) of molecule captioning task on the CheBI-20 dataset. **Bold** indicates the best performance and underline indicates the second best performance.

| Model | BLEU-2↑ | BLEU-4↑ | ROUGE-1↑ | ROUGE-2↑ | ROUGE-L↑ | METEOR↑ |
|---|---|---|---|---|---|---|
| T5-Small [38] | 50.1 | 41.5 | 60.2 | 44.6 | 54.5 | 53.2 |
| T5-Base [38] | 51.1 | 42.3 | 60.7 | 45.1 | 55.0 | 53.9 |
| T5-Large [38] | 55.8 | 46.7 | 63.0 | 47.8 | 56.9 | 58.6 |
| MolT5-Small (T5-Small) [11] | 51.9 | 43.6 | 62.0 | 46.9 | 56.3 | 55.1 |
| MolT5-Base (T5-Base) [11] | 54.0 | 45.7 | 63.4 | 48.5 | 57.8 | 56.9 |
| MolT5-Large (T5-Large) [11] | 59.4 | 50.8 | 65.4 | 51.0 | 59.4 | 61.4 |
| MoMu-Small (T5-Small) [39] | 53.2 | 44.5 | - | - | 56.4 | 55.7 |
| MoMu-Base (T5-Base) [39] | 54.9 | 46.2 | - | - | 57.5 | 57.6 |
| MoMu-Large (T5-Large) [39] | 59.9 | 51.5 | - | - | 59.3 | 59.7 |
| InstructMol (Vicuna-7B) [6] | 47.5 | 37.1 | 56.6 | 39.4 | 50.2 | 50.9 |
| MolCA (OPT-125M) [33] | 61.6 | 52.9 | 67.4 | 53.3 | 61.5 | 63.9 |
| MolCA (OPT-1.3B) [33] | 63.9 | 55.5 | 69.7 | 55.8 | 63.6 | 66.9 |
| UniMoT (LLaMA2-7B) | **66.4** | **58.3** | **72.2** | **58.4** | **66.4** | **70.3** |

- Caption-guided molecule generation involves creating molecular structures from textual descriptions, leveraging NLP and cheminformatics to interpret and translate descriptions into chemical structures.

- Reagent prediction focuses on identifying suitable reagents for given reactants and desired products, optimizing synthetic routes.

- Forward reaction prediction forecasts probable products from specific reactants and reagents, using knowledge of chemical reactivity.

- Retrosynthesis deconstructs target molecules into simpler starting materials.

In molecule generation tasks, evaluating the quality of generated molecules involves several metrics that measure different aspects of similarity and validity.

- Exact Match checks if the generated molecule is identical to the target molecule, offering a stringent criterion for precise replication but potentially overlooking chemically similar variants.

- The BLEU score [37], adapted from machine translation, measures the overlap of n-grams (short sequences of atoms or bonds) between generated and target molecules, thus assessing partial similarities.

- Levenshtein Distance [22] evaluates the minimum number of edits needed to transform the generated molecule into the target, providing insight into structural changes required.

- RDKit [20], MACCS [10], and Morgan [36] Fingerprint Similarities compare the generated and target molecules based on various molecular fingerprinting methods, which capture different aspects of molecular structure and properties.

- The Validity [19] metric assesses the proportion of chemically valid molecules generated, ensuring that the output consists of plausible chemical structures.

Together, these metrics offer a comprehensive evaluation framework, balancing exact matches with structural and chemical validity.

# E   Limitations

While UniMoT demonstrates considerable advancements in unifying molecule and text modalities for comprehensive understanding and generation tasks, several limitations must be acknowledged. Although UniMoT exhibits strong performance in molecule-to-text and text-to-molecule tasks, it has not been extensively tested on more complex molecule generation tasks such as molecule editing, which require precise modifications to molecular structures. Future work could explore extending UniMoT's capabilities to handle such sophisticated molecular manipulations.

Due to the scarcity of annotated data in the molecular field, the training of UniMoT is less extensive compared to fields like computer vision. This limitation restricts the model's ability to fully learn and generalize from diverse molecular structures and properties. In contrast, the visual domain benefits

Table 12: Examples of molecule captioning task on the ChEBI-20 dataset. We highlight in blue the text that accurately describes the molecule structures in the generated caption, ensuring alignment with the ground truth.

| Molecule | Generated Molecule Caption | Ground Truth |
|---|---|---|
|  | The molecule is an optically active form of phenylalaninate having D-configuration. It is a conjugate base of a D-phenylalanine. It is an enantiomer of a L-phenylalaninate. | The molecule is the D-enantiomer of phenylalaninate. It is a conjugate base of a D-phenylalanine. It is an enantiomer of a L-phenylalaninate. |
|  | The molecule is an ammonium ion that is the conjugate acid of 2-phenylpropylamine arising from protonation of the primary amino function; major species at pH 7.3. It has a role as a human metabolite, an Escherichia coli metabolite and a mouse metabolite. It is a conjugate acid of a 2-phenylpropylamine. | The molecule is the cation obtained by protonation of the amino group of 2-phenylethylamine. It has a role as a human metabolite and an Escherichia coli metabolite. It is a conjugate acid of a 2-phenylethylamine. |
|  | The molecule is an enamide obtained by the carboxy group of trans-cinnamic acid with the secondary amino group of (2S,5R)-1,2,5-trimethylpiperazine. It has a role as an Aspergillus metabolite. It is an alkaloid, a N-acylpiperazine, an enamide and a tertiary carboxamide. It derives from a trans-cinnamic acid. | The molecule is an enamide obtained by formal condensation of the carboxy group of trans-cinnamic acid with the secondary amino group of (2R,5R)-1,2,5-trimethylpiperazine. It has a role as an Aspergillus metabolite. It is a N-acylpiperazine, a N-alkylpiperazine, an alkaloid, an enamide and a tertiary carboxamide. It derives from a trans-cinnamic acid. |
|  | The molecule is an (omega-1)-hydroxy fatty acid ascaroside obtained by formal condensation of the alcoholic hydroxy group of (10R)-10-hydroxylauric acid with ascarylopyranose (the alpha anomer). It is a metabolite of the nematode Caenorhabditis elegans. It has a role as a Caenorhabditis elegans metabolite. It is a monocarboxylic acid and an (omega-1)-hydroxy fatty acid ascaroside. It derives from an (11R)-11-hydroxylauric acid. It is a conjugate acid of an ascr18(1-). | The molecule is an (omega-1)-hydroxy fatty acid ascaroside obtained by formal condensation of the alcoholic hydroxy group of (10R)-10-hydroxyundecanoic acid with ascarylopyranose (the alpha anomer). It is a metabolite of the nematode Caenorhabditis elegans. It is a monocarboxylic acid and an (omega-1)-hydroxy fatty acid ascaroside. It derives from a (10R)-10-hydroxyundecanoic acid. It is a conjugate acid of an ascrblue18(1-). |
|  | The molecule is a 2-oxo monocarboxylic acid that is pyruvic acid in which one of the methyl hydrogens is substituted by a 4-vinylcyclohex-2-en-1-yl group. It has a role as a plant metabolite. It derives from a pyruvic acid. It is a conjugate acid of a 4-[(1E)-4-vinylcyclohex-2-en-1-yl]pyruvate. | The molecule is a 2-oxo monocarboxylic acid that is pyruvic acid in which one of the methyl hydrogens has been replaced by a methylenecyclopropyl group. It has a role as a rat metabolite and a xenobiotic metabolite. It is a 2-oxo monocarboxylic acid, a member of cyclopropanes and an olefinic compound. It derives from a pyruvic acid. |

Table 13: Accuracy (%) of molecule-text retrieval task on the PCdes and MoMu datasets. **Bold** indicates the best performance and underline indicates the second best performance. We report the performance of retrieval using a batch of 64 random samples and the entire test set.

(a) Accuracy (%) of molecule-text retrieval task on the PCdes dataset.

| Model | Retrieval in batch | | Retrieval in test set | |
|---|---|---|---|---|
| | M2T (%) | T2M (%) | M2T (%) | T2M (%) |
| Sci-BERT [2] | 62.6 | 61.8 | 60.7 | 60.8 |
| KV-PLM [58] | 77.9 | 65.0 | 75.9 | 64.3 |
| MoMu (Sci-BERT) [39] | 80.6 | 77.0 | 79.1 | 75.5 |
| MoMu (KV-PLM) [39] | 81.1 | 80.2 | 80.2 | 79.0 |
| MoleculeSTM [31] | 86.2 | 83.9 | 84.6 | 85.1 |
| MolCA (OPT-1.3B) [33] | 91.4 | 88.4 | 90.5 | 87.6 |
| 3D-MoLM (LLaMA2-7B) [25] | 92.3 | **89.6** | 91.2 | **88.5** |
| UniMoT (LLaMA2-7B) | **92.6** | 89.4 | **91.6** | 88.3 |

(b) Accuracy (%) of molecule-text retrieval task on the MoMu dataset.

| Model | Retrieval in batch | | Retrieval in test set | |
|---|---|---|---|---|
| | M2T (%) | T2M (%) | M2T (%) | T2M (%) |
| Sci-BERT [2] | 1.4 | 1.6 | 0.3 | 0.3 |
| KV-PLM [58] | 1.5 | 1.3 | 0.5 | 0.3 |
| MoMu (Sci-BERT) [39] | 45.7 | 40.0 | 43.3 | 43.4 |
| MoMu (KV-PLM) [39] | 46.2 | 38.5 | 43.7 | 43.5 |
| MoleculeSTM [31] | 81.8 | 81.9 | 75.8 | 74.5 |
| MolCA (OPT-1.3B) [33] | 83.7 | 84.3 | 88.6 | 87.3 |
| 3D-MoLM (LLaMA2-7B) [25] | 84.9 | 85.4 | 89.9 | 88.7 |
| UniMoT (LLaMA2-7B) | **85.4** | **85.6** | **90.3** | **89.0** |

648 from abundant labeled datasets, allowing for more comprehensive training and better performance.
649 Addressing this data scarcity in the molecular domain is crucial for improving UniMoT's training
650 effectiveness and overall capabilities.

651 The current empirical evaluations, though extensive, are primarily conducted on standard datasets
652 and benchmarks; expanding the evaluation to a broader array of datasets and real-world scenarios
653 will provide a more comprehensive understanding of the model's robustness and generalizability.

654 # F   Broader Impacts

655 The development of UniMoT, a unified model for molecule and text modalities, has significant
656 potential to positively impact various fields. UniMoT can streamline the drug discovery process by
657 enabling efficient molecule generation and optimization based on textual descriptions. In material
658 science, it can aid in discovering new materials with desirable properties. Additionally, UniMoT
659 can enhance research collaboration between chemists, biologists, and data scientists by integrating
660 molecular and textual data, leading to comprehensive research insights and innovative solutions.

661 This paper does not pose any ethical concerns. The study does not involve human subjects and follows
662 proper procedures for data set releases. There are no potentially harmful insights, methodologies, or
663 applications. Additionally, there are no conflicts of interest or sponsorship concerns. Discrimination,
664 bias, and fairness issues are not applicable. Privacy and security matters have been appropriately
665 addressed, legal compliance has been maintained, and research integrity has been upheld.

