# OpenReview forum: "UniMoT: Unified Molecule-Text Language Model with Discrete Token Representation"
_NeurIPS.cc/2024/Conference — Submitted to NeurIPS 2024_

### Official Review · Reviewer_pDSz · 2024-06-19

**Soundness:** 2
**Presentation:** 3
**Contribution:** 1
**Rating:** 2
**Confidence:** 5

**Summary:**

The authors propsed Uni-MoT, a unified structure to align molecules with texts with a VQ tokenizer and the Q-Former in BLIP-2. By treating molecules as new word tokens in the codebook, Uni-MoT aligns the discrete token representation for molecules and texts, while also following the autoregressive manner of LLMs. The training of Uni-MoT follows four main stages, Causal Q-Former Pretraining, Molecule Tokenizer Pretraining, Unified Molecule-Text Pretraining, Task-Specific Instruction Tuning. The experiments demonstrate that Uni-MoT can achieve better performance compared to the selected baselines.

**Strengths:**

1. The performance of Uni-MoT is overall good and better than the baseline models.
2. Uni-MoT provides a alternative solution for aliging molecules with texts.

**Weaknesses:**

1. Although the authors claim that Uni-MoT follows a different structure as shown in Figure 1 c, it turns out that it still follows the BLIP-2 [1] structure, which has been widely used to align 2D molecule graph [2] and 3D molecule information [3]. Thus, **the technical contribution and novelty of this paper are extremely limited**. Especially when VQ-VAE [4] is also a well-developed structure adopted in computer vision. This paper is more like simply swapping the input of the Q-former in BLIP-2.
2. The experiments are only conducted on a single serie of LLM, Llama-2, **which is not sufficient to demonstrate the model agnosticsm of Uni-MoT**. In fact, LLMs like Mistral [5] and Meditron [6] might possibly achieve a better performance. Meanwhile, the selection of Llama-2 is also not convincing, as Llama-2 is not specially pre-trained on chemistry or biomedical corpora.
3. **The seletion of the datasets is also worth discussion**. The result on ChEBI-20 is presented in Appendix, while the main experiments are conducted on PubChem. I am wondering why not also conduct the remaining experiments on ChEBI-20, as the data scale of ChEBI-20 is much larger than PubChem. At the same time, the reverse task, text-based molecule generation, on ChEBI-20 and PubChem is surprisingly not presented.
4. **The comparison with the baselines is not fair enough**. For example, MolCA adopts Galactica-1.3B as its backbone model, which is much smaller than Llama-2-7B. Thus, the proposed method can not demonstrate its superiority compared to previous methods. Notably, since the authors mentioned MolCA and 3D-MoLM, it is necessary to discuss the possible affects of the modalities. Furthermore, several SOTA baseline models like BioT5 [7] and BioT5+ [8] are not discussed.
5. **The ablation study is also not sufficient without providing the naive fine-tuning performance of Llama-2**. Besides, as Uni-MoT incorporates the VQ tokenizer, it is also important to discuss the size of the codebook.
6. During the pre-training stages, it should ensure that the **data leakage** is avoided. Considering the pre-training dataset adopted has overlaps with the fine-tuning dataset [7, 8], the performance gain could possibly come from the data leakage.
7. **Some claims and definitions are confusing**. e.g. In Line 44, "a unified molecule-text LLM", I do not agree with the claim of an "LLM". The "LLM" is still the Llama-2. It should be like "structure" or something.

#### References
[1 ]Li, J., Li, D., Savarese, S., & Hoi, S. (2023, July). Blip-2: Bootstrapping language-image pre-training with frozen image encoders and large language models. In International conference on machine learning (pp. 19730-19742). PMLR.
[2] Liu, Z., Li, S., Luo, Y., Fei, H., Cao, Y., Kawaguchi, K., ... & Chua, T. S. (2023). Molca: Molecular graph-language modeling with cross-modal projector and uni-modal adapter. arXiv preprint arXiv:2310.12798.
[3] Li, S., Liu, Z., Luo, Y., Wang, X., He, X., Kawaguchi, K., ... & Tian, Q. (2024). Towards 3D Molecule-Text Interpretation in Language Models. arXiv preprint arXiv:2401.13923.
[4] Yu, J., Li, X., Koh, J. Y., Zhang, H., Pang, R., Qin, J., ... & Wu, Y. (2021). Vector-quantized image modeling with improved vqgan. arXiv preprint arXiv:2110.04627.
[5] Jiang, A. Q., Sablayrolles, A., Mensch, A., Bamford, C., Chaplot, D. S., Casas, D. D. L., ... & Sayed, W. E. (2023). Mistral 7B. arXiv preprint arXiv:2310.06825.
[6] Chen, Z., Cano, A. H., Romanou, A., Bonnet, A., Matoba, K., Salvi, F., ... & Bosselut, A. (2023). Meditron-70b: Scaling medical pretraining for large language models. arXiv preprint arXiv:2311.16079.
[7] Pei, Q., Zhang, W., Zhu, J., Wu, K., Gao, K., Wu, L., ... & Yan, R. (2023). Biot5: Enriching cross-modal integration in biology with chemical knowledge and natural language associations. arXiv preprint arXiv:2310.07276.
[8] Pei, Q., Wu, L., Gao, K., Liang, X., Fang, Y., Zhu, J., ... & Yan, R. (2024). BioT5+: Towards Generalized Biological Understanding with IUPAC Integration and Multi-task Tuning. arXiv preprint arXiv:2402.17810.

**Questions:**

1. Could the authors conduct the experiments on more backbone LLMs? For example, on Galactica-1.3B, is the performance better than MolCA? Or on Llama-2-7B, is the performance still better than MolCA(Llama-2-7B)?
2. Could the authors justify the selection of the datasets?
3. Could the authors compare the proposed method with stronger baselines? Let's say, when the authors claim the SOTA performance, it should be noted BioT5 and BioT5+ are not included in discussion.
4. Could the authors provide more solid ablation study?

My main questions are listed above, but I do expect the authors could discuss more about their weaknesses.

**Limitations:**

Yes. They have discussed the limitations as not enough tasks, data scarcity, and real scenarios.

---

> ### Author Rebuttal · Authors · 2024-08-07
>
> > W1. Limited novelty.
>
> We would like to clarify that the Q-Former is only a part of our VQ-driven tokenizer. UniMoT does **not** follow the BLIP-2 structure in its entirety. UniMoT adopts a tokenizer-based architecture, which uses **discrete tokens**, fundamentally different from adapter-based architectures that use **continuous embeddings**, such as MolCA and 3D-MoLM. We would like to clarify the novel aspects of our work to address your concerns. Due to space constraints, please refer to the *global rebuttal* for details on the novelty of our work.
>
> > W2. Different backbone LLMs.
>
> We have extended our experiments to include additional LLMs to validate the generalizability of UniMoT. The performance is shown in Table 1 of the *global rebuttal PDF*. Our experiments show that UniMoT performs well across multiple LLMs, including Galactica and Mistral series, demonstrating its robustness and generalizability. This confirms that UniMoT can be successfully applied to other SOTA LLMs.
>
> Llama-2 was chosen initially due to its versatility and performance in handling diverse data types. While it may not be pre-trained specifically on chemistry or biomedical corpora, our pre-training and fine-tuning processes on relevant datasets make sure that the model learns the necessary domain-specific information.
>
> > W3 & Q2. Selection of datasets.
>
> The PubChem dataset is significantly larger than the ChEBI-20 dataset. This allows for more comprehensive pretraining. The table below summarizes the size of both datasets:
>
> |          | Pretrain | Train  | Valid | Test  |
> | :------- | :------- | :----- | :---- | :---- |
> | PubChem  | 301,658  | 12,000 | 1,000 | 2,000 |
> | ChEBI-20 | -        | 26,407 | 3,300 | 3,300 |
>
> The ChEBI-20 dataset replaces molecular names with “the molecule,” focusing the model on properties rather than names. However, accurately predicting molecular names demonstrates the model’s understanding of molecular structures. Thus, we conducted the main experiments on PubChem.
>
> For molecule generation tasks, we utilized the Mol-Instructions benchmark. The caption-guided molecule generation task within this benchmark actually uses the **PubChem** dataset.
>
> > W4 & Q1. Models with comparable sizes.
>
> We provide a detailed performance comparison between UniMoT and MolCA using models of comparable sizes. The performance is shown in Table 2 of the *global rebuttal PDF*. UniMoT consistently outperforms MolCA when using the same Galactica-125M, Galactica-1.3B, and Llama-2-7B backbones. This demonstrates the effectiveness of our proposed UniMoT.
>
> **For the modalities**, adapter-based architectures require the LLM to directly output **SMILES strings** to perform molecule generation tasks. This approach relies on strong alignment between SMILES strings and molecule captions during pretraining. In practice, this alignment is challenging to establish, leading to suboptimal performance in text-to-molecule generation tasks.
>
> Our UniMoT treats molecular and textual data **equally** under a unified token representation. This enables the LLM to unify the modalities under a shared autoregressive training paradigm. The **molecule tokens** encapsulate high-level molecular and textual information, providing a richer representation than SMILES strings alone.
>
> > Q3. BioT5 and BioT5+ baselines.
>
> We have included BioT5 and BioT5+ in our comparative analysis. We selected the molecule captioning task from the molecule comprehension tasks with the molecule generation tasks. The results are presented in Tables 3 and 4 of the *global rebuttal PDF*. These comparisons demonstrate the comparable performance of our model with BioT5 and BioT5+ in comprehension and generation tasks. In future revisions, we will incorporate performances of BioT5 and BioT5+ to provide a more comprehensive comparison.
>
> > W5 & Q4. Ablation studies on fine-tuning strategy and codebook size.
>
> We would like to highlight that we have indeed provided the performance of full fine-tuning of Llama-2-7B in Table 6 of our manuscript. The pre-trained Llama-2-7B model was directly fine-tuned on the task-specific dataset without any specialized techniques.
>
> The choice of 2048 for the molecule codebook size is based on a balance between **model complexity** and **performance**. A larger codebook could potentially capture more subtle interactions between molecules and text. However, there may be some codes that are not often used on large codebooks. A smaller codebook may result in nearby embeddings being assigned the same code, which reduces the granularity of the representation. The performance with different codebook sizes on the molecule captioning task is shown in Table 5 of the *global rebuttal PDF*. The results demonstrate that the codebook size of 2048 consistently provides the best performance.
>
> > W6. Data leakage.
>
> We appreciate the concern regarding potential data leakage. We take steps to ensure that data leakage is avoided throughout our training process. We list the datasets employed at each stage:
>
> - Stage 1: Causal Q-Former Pretraining
>   - Datasets: PubChem pretrain subset
> - Stage 2: Molecule Tokenizer Pretraining
>   - Datasets: PubChem pretrain subset, CheBI-20 train subset
> - Stage 3: Unified Molecule-Text Pretraining
>   - Datasets: PubChem pretrain subset, CheBI-20 train subset
> - Stage 4: Task-Specific Instruction Tuning
>   - Datasets: PubChem, PCdes, MoMu, MoleculeNet, QM9, USPTO (all train subsets)
>
> >W7. The terminology.
>
> Thank you for your feedback regarding the terminology. When we refer to “a unified molecule-text LLM,” we mean that the LLM can process input sequences that may be purely molecular, purely textual, or multi-modal (combining both molecules and text). This capability is achieved through treating molecular and textual data equally under a unified token representation. The LLM learns to predict the next token in a sequence, regardless of whether the sequence consists of molecules, text, or a combination of both.

---

> ### Comment · Reviewer_pDSz · 2024-08-07
> **Reviewer Response**
>
> Thanks for the clarification and extended experiments.
> * W1: As the author explained, the model structure is exactly similar to MolCA and 3D-MoLM. In this case, the novelty should be further discussed and I personally don’t see enough novelty in the model structure.
> * W3: I do not agree with the statement of the dataset size. The training part of PubChem is only 12000 samples, while ChEBI-20 is even double. The performance comparison is unfair because you introduce more related data.
> * W6: Still not guarantee that the data will not leak during the pretraining stages.
> * W7: I don’t think so.

---

> ### Author Response · Authors · 2024-08-09
> **Response to Reviewer pDSz (1/2)**
>
> Thank you for your thorough review and valuable feedback. Below, we address the further concerns you have raised in detail.
>
> > I personally don’t see enough novelty in the model structure.
>
> We respectfully disagree with the assessment that the model structure is similar to MolCA and 3D-MoLM. The architecture of UniMoT is fundamentally different from these models. Specifically:
>
> 1. **Model Architecture**:
>
>    - **MolCA and 3D-MoLM**: These models follow the **Q-Former** architecture, as depicted in Figure 1(b) of our paper. In this architecture, the input to the LLM is molecule embeddings.
>    - **UniMoT**: Our model employs a **VQ-VAE** architecture, as illustrated in Figure 1(c). The key difference lies in the input to the LLM: UniMoT uses molecule tokens instead of embeddings.
>
> 2. **Supervision Signal for Molecule Modality**:
>
>    - **MolCA and 3D-MoLM**: These models do not incorporate any supervision signal for the molecule modality. This limitation significantly constrains their capacity and effectiveness.
>    - **UniMoT**: In contrast, UniMoT provides direct supervision for the molecule tokens through autoregressive pretraining. This allows for stronger generation abilities and enables the molecule tokens to be decoded into molecules.
>
> 3. **Text-to-Molecule Generation**:
>
>    - **MolCA and 3D-MoLM**: These models rely solely on SMILES strings for text-to-molecule generation tasks. This heavily relies on strong alignment between SMILES strings and molecule captions during pretraining. In practice, achieving this alignment is challenging, leading to suboptimal performance in text-to-molecule generation tasks.
>    - **UniMoT**: Our model not only supports text-to-molecule generation but also benefits from the direct supervision of molecule tokens, leading to improved accuracy in generation tasks. The molecule tokens encapsulate high-level molecular and textual information, providing a richer representation than SMILES strings alone.
>
> 4. **Cross-Modal Alignment**:
>
>    - **MolCA and 3D-MoLM**: These models rely heavily on the Q-Former for cross-modal alignment, which can be restrictive in certain scenarios.
>
>    - **UniMoT**: Our approach allows for the Q-Former to be **discarded entirely**. Instead, we quantize the molecule features into molecule tokens using the VQ-VAE pipeline, with a codebook size of 1024. We report the performance of the molecule captioning task on the PubChem dataset. This demonstrates that **the performance remains comparable even without the Causal Q-Former.**
>
>      |                            | BLEU-2 | BLEU-4 | ROUGE-1 | ROUGE-2 | ROUGE-L | METEOR |
>      | -------------------------- | :----- | :----- | :------ | :------ | :------ | :----- |
>      | UniMoT w/o Causal Q-Former | 28.1   | 20.8   | 33.2    | 20.7    | 30.1    | 30.8   |
>      | UniMoT w/ Causal Q-Former  | 31.3   | 23.8   | 37.5    | 23.7    | 33.6    | 34.8   |
>
> > I do not agree with the statement of the dataset size.
>
> - The PubChem training set is divided into two distinct subsets: a pretraining subset and a training subset. These subsets do not intersect, which ensures that data leakage is avoided. Specifically, the **pretraining subset** (301,658 molecule-text pairs) is used exclusively for pretraining the model. And the **training subset** (12,000 molecule-text pairs) is reserved for instruction tuning.
>
> - ChEBI-20 only has a single training set, which consists of 26,407 pairs. This set is used solely for pretraining.
>
> When comparing dataset sizes, it is important to consider the relevant subsets. The PubChem pretraining subset (301,658 pairs) is significantly larger than the ChEBI-20 training set (26,407 pairs). This is why we chose to use PubChem for pretraining, following a practice as used in InstructMol, MolCA, and 3D-MoLM.
>
> > The performance comparison is unfair because you introduce more related data.
>
> We would like to clarify why the performance comparison is indeed fair.
>
> The ChEBI-20 dataset replaces molecular names with “the molecule,” which shifts the model's focus toward predicting molecular properties rather than specific names. However, accurately predicting molecular names is crucial for demonstrating a model’s understanding of molecular structures. To address this, we conducted experiments on both PubChem and ChEBI-20, and the results for **ChEBI-20** are presented **in the Appendix**, while the main experiments on **PubChem** are included **in the main text**.
>
> All baseline models were evaluated on the same **PubChem test set** on the experiments in the main text. Since recent baselines like MolCA, 3D-MoLM, and InstructMol also use PubChem for pretraining, our comparison is consistent and fair.

---

> ### Author Response · Authors · 2024-08-09
> **Response to Reviewer pDSz (2/2)**
>
> > Still not guarantee that the data will not leak during the pretraining stages.
>
> We can assure that data leakage is effectively prevented during the pretraining and instruction tuning stages. Here's how we ensure this:
>
> - The PubChem dataset is carefully divided into two distinct subsets. The **pretraining subset** is exclusively used during the pretraining stages. And the **training subset** is reserved for instruction tuning. These subsets do not intersect, ensuring that data used for instruction tuning is entirely separate from the data used during pretraining.
>
> - The additional datasets used during the instruction tuning stage (PCdes, MoMu, MoleculeNet, QM9, USPTO) are also separate and distinct from the datasets used in the pretraining stages. These datasets do not appear in the pretraining phase, ensuring that there is no overlap or data leakage.
>
> Thank you for your interest in our work. Your insightful comments have contributed to strengthening our paper. Should you have any questions or require additional information, please do not hesitate to contact us.

---

> ### Comment · Reviewer_pDSz · 2024-08-13
> **Reviewer Response**
>
> Thanks to the authors for explaining my concerns.
> #### However, I hope to clarify my views again considering the model structure.
> 1. The model structure and model components proposed in this paper have already been used in the field of Computer Vision. The authors brought nothing new. In other words, the authors assemble the Q-Former with the VQ-VAE.
> 2. Although the authors claim they have better "cross-modal alignment", the benefits are not satisfying at all. We can call it a small improvement, but I don't see enough potential for this work to further benefit the community, as well as the development of computational biology. Frankly speaking, this work still does not align molecular graphs well with texts, and it can not help chemists save their workload.
>
> #### The dataset size
> I never see anyone compare their pretraining dataset with the downstream training set.
>
> #### Unfair comparison
> The responses did not address the problem well. Let's change the question. When comparing the downstream performance between Mistral-7B and Llama-2-7B, Mistral-7B shows better performance. Then we could say that the pre-training of Mistral-7B is better. However, when both the two models are adopted in your model structure, is it still fair to say it is *your model structure* that brings the improvement? No. The performance gain is partly from pre-training. When you compare your method with the previous baselines, how could you ensure that your model structure is better?
>
> #### data leakage
> I don't think the authors answered my question. Yes, the datasets are not used in pertaining. However, there are molecules in downstream tasks that appear in your pre-training set. When you pre-train the model, the information has already been leaked. It is worth noting that in the work of BioT5, the authors did a special operation that removed all the overlapping molecules from the pre-training set.
>
>
> Overall, I strongly believe this work is not qualified to be accepted, and these issues should be further addressed. I will maintain my score.

---

### Official Review · Reviewer_d9hA · 2024-07-09

**Soundness:** 2
**Presentation:** 2
**Contribution:** 2
**Rating:** 4
**Confidence:** 4

**Summary:**

This work presents a new molecule LLM that uses a pretrained tokenizer to replace the projection layer. The tokenizer consists a Q-Former and a VQ module, which are trained with consistency loss. The model is evaluated on molecular understanding and generation tasks.

**Strengths:**

- The authors provided a novel framework to align text representation in LLMs and the molecules.
- The authors conducted experiments on many datasets.

**Weaknesses:**

- The claim that adapter-based methods cannot do text-to-molecule generation tasks in not accurate (Sec.1). They can always adapt text-encoder to a pre-trained SMILES or graph generator. This may make this work not well motivated.
- In the tokenizer (Fig.2), it seems that multiple alignment methods are required to train the model, including: (1)the molecule and text contrastive in Q-Former, (2), the aligning MSE loss, (3) the SMILES decoder reconstruction. It's not clear about the design reasons, and if all of them are required.
- Given the learnable query has a fixed size, it may not perform well for larger molecules.

**Questions:**

- In Sec 3.3, what are the data that used in different stages?
- In Fig. 1, the model can output both molecule and text at the same time. But in Fig. 3, seems only text or molecule get loss for back propagation. Is this consistent and can authors share more training details?
- In all experiments, does this model pretrained with same data as all baselines?

**Limitations:**

Yes

---

> ### Author Rebuttal · Authors · 2024-08-07
>
> > W1. Text-to-molecule generation tasks.
>
> As Reviewer d9hA pointed out, adapter-based architectures can also perform text-to-molecule generation tasks. We will revise the manuscript accordingly. However, our contention is that such methods typically do not perform as well as our approach, as demonstrated in Table 4 of our paper. Here, we elaborate on the key differences and advantages of tokenizer-based architecture over adapter-based architectures:
>
> - **Limitations of Adapter-Based Architectures**:  Adapter-based architectures require the LLM to directly output **SMILES strings** to perform molecule generation tasks. This approach relies heavily on strong alignment between SMILES strings and molecule captions during pretraining. In practice, this alignment is challenging to establish, leading to suboptimal performance in text-to-molecule generation tasks.
>
> - **Advantages of Tokenizer-Based Architecture**: Our method leverages a tokenizer to convert molecular features and text captions into **molecule tokens**. These tokens encapsulate high-level molecular and textual information, providing a richer representation than SMILES strings alone. By linking molecule tokens generated by the tokenizer to the molecule captions during molecule-to-text and text-to-molecule pretraining, our approach ensures that the model learns to understand and generate molecule tokens in an autoregressive manner.
>
> > W2. Alignment methods.
>
> We outline the purpose and necessity of each alignment method below:
>
> - **Molecule-Text Contrastive Learning**: Molecule-Text Contrastive Learning is employed solely in Stage 1: Causal Q-Former Pretraining. The primary objective here is to align the molecule features with the text captions, ensuring that the queries incorporate aligned molecular and textual information. This contrastive learning helps the Causal Q-Former to understand and relate molecular structures with their corresponding textual descriptions.
>
> - **Aligning MSE Loss**: The aligning MSE loss is used exclusively in Stage 2: Molecule Tokenizer Pretraining. Its purpose is to align the discrete latent space of molecule tokens with the continuous latent space of a molecular generative model. The alignment ensures that the tokenized representation can be mapped to the decoder's latent space.
>
> - **SMILES Decoder Reconstruction**: The reconstruction is primarily relevant during inference. During training, we focus on obtaining the reconstruction loss (the aligning MSE loss) to perform back-propagation. The actual reconstruction of the molecule via the SMILES decoder is done during inference to generate the final molecular output.
>
> > W3. Fixed size of queries.
>
> The design of our Causal Q-Former effectively addresses the concern regarding the fixed size of the queries through attention mechanisms:
>
> - **Self-Attention Mechanism**: Although the query size is fixed, the Causal Q-Former employs a dynamic self-attention mechanism for the queries. This allows the model to adaptively capture molecular details necessary for understanding and generating textual descriptions.
>
> - **Cross-Attention Mechanism**: The Causal Q-Former also utilizes a cross-attention mechanism where the queries selectively attend to different aspects of the molecule features. This allows the model to adaptively focus on different parts of the molecule based on its complexity and size.
>
> We also conducted an ablation study to evaluate the performance of UniMoT with different query sizes, as presented in Table 6 of the *global rebuttal PDF*. The results show that increasing the query size improves performance, with the best results observed at a query size of 32. However, since a query size of 32 requires significantly more training time and memory, we still opt to use a query size of 16.
>
> > Q1. Datasets usage.
>
> Below, we provide a list of the specific datasets employed at each stage:
>
> - Stage 1: Causal Q-Former Pretraining
>   - Datasets: PubChem pretrain subset
> - Stage 2: Molecule Tokenizer Pretraining
>   - Datasets: PubChem pretrain subset, CheBI-20 train subset
> - Stage 3: Unified Molecule-Text Pretraining
>   - Datasets: PubChem pretrain subset, CheBI-20 train subset
> - Stage 4: Task-Specific Instruction Tuning
>   - Datasets: PubChem, PCdes, MoMu, MoleculeNet, QM9, USPTO (all train subsets)
>
> > Q2. Different outputs in Figures 1 and 3.
>
> Yes, it is consistent. Figure 1 is designed to convey the core idea of our method. It shows the **autoregressive prediction** of molecule and text tokens. When data is organized as **interleaved** molecule and text tokens, the model can predict the next token (whether it is a molecule or text) indiscriminately. Each token prediction is supervised, and learning occurs across both types of tokens.
>
> Figure 3 illustrates the **separate supervision** of molecule and text tokens adopted in Stage-3, which is the practical implementation of our training process. Instead of using an interleaved organization of molecule and text tokens, we use the dataset as provided, consisting of molecule-text pairs. This involves two tasks:
>
> - Molecule-to-text autoregression: Using molecule tokens as a prompt to supervise and generate text.
>
> - Text-to-molecule autoregression: Using text as a prompt to supervise and generate molecule tokens.
>
> We utilize the PubChem and CheBI-20 datasets for the unified molecule-text pretraining.
>
> > Q3. Datasets with baselines.
>
> We provide a list of the datasets used for pretraining and fine-tuning UniMoT and the baselines:
>
> - UniMoT:
>
>   - Pretraining: PubChem and CheBI-20
>
>   - Fine-Tuning: Specific task datasets
>
> - MolT5:
>
>   - Pretraining: Colossal Clean Crawled Corpus (C4) and ZINC
>
>   - Fine-Tuning: CheBI-20
>
> - InstructMol:
>
>   - Pretraining: PubChem
>
>   - Fine-Tuning: Specific task datasets
>
> - MolCA:
>
>   - Pretraining: PubChem
>
>   - Fine-Tuning: PubChem
>
> - 3D-MoLM:
>
>   - Pretraining: PubChem
>
>   - Fine-Tuning: 3D-MoIT

---

### Official Review · Reviewer_2Dfq · 2024-07-11

**Soundness:** 2
**Presentation:** 3
**Contribution:** 2
**Rating:** 5
**Confidence:** 4

**Summary:**

The authors propose to use a vector-quantized tokenizer that incorporates a Q-Former to connect pre-trained molecule encoder, SMILES encoder, and SMILES decoder so that a language model can integrate molecule and text modalities. Based on the proposed tokenizer, the authors introduce a four-stage training strategy to train UniMoT, a unified molecule-text LLM proposed in this submission. The performance of the UniMoT is evaluated empirically with 7 tasks in the areas of molecule comprehension and molecule generation. Some ablation studies are also conducted.

**Strengths:**

- This paper is well-organized and well-written.
- Using a vector-quantized tokenizer and a Q-Former to connect different modalities could be somewhat novel.
- The proposed UniMoT outperforms baselines in most cases and reach comparable performances in others.
- The authors provide many implement details which increases the reproducibility.

**Weaknesses:**

- All the components are borrowed from existing works. Besides the pre-trained molecule encoder, SMILES encoder and decoder, both vector quantization and Q-Former are proposed in previous works [1][2]. The Q-Former part of this paper (Appendix A) is very similar to Q-Former's original paper. Even Figure 4 in this paper is very similar to Figure 2 in Q-Former's original paper [2].
- When generating molecules the proposed method relies heavily on a pre-trained decoder. In the decoder's original paper, the reported validity is 99.9 and no guarantee is provided that the generated SMILES string will be always valid [3]. The 100% validity reported in this paper could be attributed to overfitting.
- Some hyperparameter choices are not well justified and studied. For instance, the molecule codebook size is set to 2048, but there is no explanation why 2048 is chosen. How the molecule codebook size affects the performance is also not studied.
- There are some existing works about molecule tokenizers [4][5], the paper lacks the comparison of the performance using different tokenizers.
- The robustness of the model is not studied. Each molecule has many synonyms, how the proposed method performs given different synonyms of the same molecule is desired to know.

[1] Van Den Oord, Aaron, and Oriol Vinyals. "Neural discrete representation learning." Advances in neural information processing systems 30 (2017).

[2] Li, Junnan, et al. "Blip-2: Bootstrapping language-image pre-training with frozen image encoders and large language models." International conference on machine learning. PMLR, 2023.

[3] Irwin, R., Dimitriadis, S., He, J., Bjerrum, E.J., 2021. Chemformer: A Pre-Trained Transformer for Computational Chemistry. Mach. Learn. Sci. Technol.

[4] Li, Xinhao, and Denis Fourches. "SMILES pair encoding: a data-driven substructure tokenization algorithm for deep learning." Journal of chemical information and modeling 61.4 (2021): 1560-1569.

[5] Schwaller, Philippe, et al. "Molecular transformer: a model for uncertainty-calibrated chemical reaction prediction." ACS central science 5.9 (2019): 1572-1583.

**Questions:**

- How do you guarantee that the output SMILES string will be 100% valid?
- Why the molecule codebook size is chosen as 2048? How will the performance change when the codebook size is changed?
- If you use other molecule tokenizers, how will the performance change?
- When pre-training with PubChem dataset, what is the input text for each molecule since there are many entries and synonyms for each molecule?
- How do you solve the problem that a molecule has many synonyms? Will your method output different answers if different synonyms of the same molecule are used in the prompt?

**Limitations:**

The authors have adequately addressed the limitations.

---

> ### Author Rebuttal · Authors · 2024-08-07
>
> > W1. Limited novelty.
>
> While we use components from existing works like Q-Former, the molecule encoder, and the SMILES encoder and decoder, we adopt a tokenizer-based architecture that uses **discrete tokens**. This is fundamentally different from adapter-based architectures that use **continuous embeddings**. Figure 4 is directly adapted from Figure 2 of Q-Former's original paper to illustrate related concepts and objectives of Causal Q-Former. We would like to clarify the novel aspects of our work to address your concerns. Due to space constraints, please refer to the *global rebuttal* for details on the novelty of our work.
>
> > W2 & Q1. The validity of generated molecules.
>
> Our model is pre-trained on extensive datasets of valid molecules, such as PubChem, which contain 0.3M of structurally diverse and chemically valid SMILES strings. This large-scale pre-training helps the model learn a robust representation of valid molecular structures. After the initial pre-training, we fine-tune our model on datasets composed exclusively of valid molecules. This fine-tuning process reinforces the learned characteristics of valid SMILES strings.
>
> Our empirical validation shows that while a few invalid molecules are generated when testing, these occurrences are rare and do not affect the overall validity rate.
>
> > W3 & Q2. Ablation study on codebook size and query size.
>
> The choice of 2048 for the molecule codebook size is based on a balance between **model complexity** and **performance**. A larger codebook could potentially capture more subtle interactions between molecules and text. However, there may be some codes that are not often used on large codebooks. A smaller codebook may result in nearby embeddings being assigned the same code, which reduces the granularity of the representation.
>
> We conducted experiments with different codebook sizes and report the performance of the molecule captioning task on the Pubchem dataset. The performance with different codebook sizes is shown in Table 5 of the *global rebuttal PDF*. The results demonstrate that the codebook size of 2048 consistently provides the best performance.
>
> We also conducted an ablation study to evaluate the performance of UniMoT with different query sizes, as presented in Table 6 of the *global rebuttal PDF*. The results show that increasing the query size improves performance, with the best results observed at a query size of 32. However, since a query size of 32 requires significantly more training time and memory, we still opt to use a query size of 16.
>
> > W4 & Q3. Comparison with different tokenizers.
>
> **Difference from Existing Tokenizers**:
>
> - The cited works [4] and [5] focus on **SMILES tokenizers**, which primarily tokenize the SMILES strings of molecules. While SMILES strings capture sequential information, they do not fully encapsulate the structural intricacies of molecules necessary for understanding and generation tasks.
>
> - Our **molecule tokenizer**, on the other hand, is designed to tokenize molecules into **molecule tokens**. This involves incorporating textual information and structural information embedded in molecule features. Our tokenizer is specifically designed to generate tokens that LLMs can understand and generate, similar to text tokens.
>
> We report the performance of the molecule captioning task on the Pubchem dataset. Note that in the SMILES tokenizer baselines, we do not incorporate molecule features; we only use the SMILES strings as input for the LLM.
>
> |                                       | BLEU-2   | BLEU-4   | ROUGE-1  | ROUGE-2  | ROUGE-L  | METEOR   |
> | :------------------------------------ | :------- | :------- | :------- | :------- | :------- | :------- |
> | SMILES Pair Encoding (Atom Tokenizer) | 6.7      | 3.9      | 8.1      | 3.3      | 6.1      | 7.0      |
> | Molecular Transformer                 | 7.9      | 5.0      | 9.6      | 4.5      | 7.4      | 8.4      |
> | UniMoT                                | **31.3** | **23.8** | **37.5** | **23.7** | **33.6** | **34.8** |
>
> > W5 & Q4 & Q5. The synonym problem.
>
> For our pretraining, we utilize the molecule-text pairs as provided by the PubChem dataset without any modification.
>
> - **Molecule-to-Text Autoregression**:
>
>   - For molecule-to-text autoregression, our model uses the canonical SMILES representation, which uniquely defines the chemical structure of a molecule.
>
>   - We also incorporate contextual tokens derived from contextual molecule embeddings. These tokens help capture the meaning of molecule names within their specific context, allowing the model to understand that different names can refer to the same chemical entity.
>
> - **Text-to-Molecule Autoregression**:
>
>   - For text-to-molecule autoregression, our model generates molecule tokens that have been aligned with textual descriptions during pretraining.
>
>   - We conducted several tests, which showed that the model's output remains consistent regardless of the synonyms used in the input. For instance, when we replace “Acetylcarnitine” with “L-Carnitine, acetyl ester” or “O-Acetyl-L-carnitine,” the output remains consistent.
>   - While the model generally produces consistent outputs regardless of the synonym used, slight variations may occur due to differences in how the model processes each synonym.

---

> > ### Comment · Reviewer_2Dfq · 2024-08-13
> >
> > Thank you for the detailed response. Most of my concerns are clarified, I will update my score accordingly.

---

> > > ### Author Response · Authors · 2024-08-13
> > >
> > > Thank you for taking the time to review our submission and for your thoughtful comments. We're glad we could address most of your concerns. We appreciate your consideration in updating the score. Please feel free to reach out if there are any further questions or points to discuss.

---

### Official Review · Reviewer_dTUx · 2024-07-12

**Soundness:** 3
**Presentation:** 3
**Contribution:** 3
**Rating:** 6
**Confidence:** 2

**Summary:**

introduce a molecule tokenizer based on Codebooks which gets integrated into UniMoT, a unified molecule-text LLM

**Strengths:**

strong performance on a variety of benchmarks; outperforms/on pair with 3D-MoLM

**Weaknesses:**

- Table 1: include CLAMP [1] as well as standard molecular fingerprints (incl. in said reference); further KV-PLM linear probing for ToxCast results in 66.3 AUROC compared to 55.03 in your paper

[1] Seidl, P., Vall, A., Hochreiter, S., & Klambauer, G. (2023, July). Enhancing activity prediction models in drug discovery with the ability to understand human language. In International Conference on Machine Learning (pp. 30458-30490). PMLR.

**Questions:**

what is the reconstruction rate of the SMILES decoder?

**Limitations:**

-

---

> ### Author Rebuttal · Authors · 2024-08-07
>
> > W1. Including CLAMP and fingerprints baselines.
>
> Thank you for your feedback and for suggesting additional baselines such as CLAMP and standard molecular fingerprints on the molecular property prediction task. We will include these baselines in our revised manuscript to provide a more comprehensive evaluation. We will also update the ToxCast results to reflect the performance of KV-PLM linear probing.
>
> > Q1. The reconstruction rate of the SMILES decoder.
>
> We measured the reconstruction rate on a test set of SMILES strings. The reconstruction rate is defined as the percentage of SMILES strings that are exactly reproduced by the decoder from their encoded representations. Our experiments show that the SMILES decoder achieves a reconstruction rate of 96% on the PubChem test set of 2,000 valid molecules.

---

### Author Rebuttal · Authors · 2024-08-07

We appreciate the reviewers’ thoughtful and constructive feedback on our manuscript. We have carefully considered each point raised and provide the following detailed clarifications:

**Novelty and technical contribution of UniMoT.**

While we use components from existing works like Q-Former, the molecule encoder, and the SMILES encoder and decoder, we adopt a tokenizer-based architecture that uses **discrete tokens**. This is fundamentally different from adapter-based architectures that use **continuous embeddings**. The novel contributions are highlighted below:

- **First Work on Molecule Tokenizer for LLMs**: To the best of our knowledge, our tokenizer is the **first** to enable the tokenization of molecular data specifically for LLMs, treating molecular and textual data equally under a unified token representation. This approach enables the LLM to handle molecular data as a foreign language, thus unifying the modalities under a shared autoregressive training paradigm.

- **Unified Framework for Molecule and Text**: Previous works often employed **adapter-based architectures** or **separate processing pipelines** for different data modalities. In contrast, our model adopts the **tokenizer-based architecture** which tokenizes molecules into discrete token sequences that the LLM can process alongside text tokens.

- **Adaptation of Q-Former with Causal Masks**: While Q-Former has been proposed in previous works, our Causal Q-Former introduces **causal masks** to generate a causal sequence of queries. This ensures compatibility with the unidirectional attention mechanism in LLMs. Moreover, the **training objectives** are tailored for Causal Q-Former and differ from those of the original Q-Former.

**Novel approach to text-to-molecule generation tasks.**

While adapter-based architectures can perform text-to-molecule generation tasks, they typically do not match the performance of our approach, as demonstrated in Table 4 of our paper. Below, we elaborate on the key differences and advantages of our tokenizer-based architecture compared to adapter-based architectures:

- **Limitations of Adapter-Based Architectures**: Adapter-based architectures require the LLM to directly output **SMILES strings** for molecule generation tasks. This approach heavily relies on strong alignment between SMILES strings and molecule captions during pretraining. In practice, achieving this alignment is challenging, leading to suboptimal performance in text-to-molecule generation tasks.
- **Advantages of Tokenizer-Based Architecture**: Our method leverages a tokenizer to convert molecular features and text captions into **molecule tokens**. These tokens encapsulate high-level molecular and textual information, providing a richer representation than SMILES strings alone. By linking molecule tokens to the molecule captions during molecule-to-text and text-to-molecule pretraining, the model learns to understand and generate molecule tokens in an autoregressive manner.

**Ablation study on different backbone LLMs.**

We have extended our experiments to include additional LLMs to validate the generalizability of UniMoT. The performance is shown in Table 1 of the *global rebuttal PDF*. Our experiments show that UniMoT performs well across multiple LLMs, including Galactica and Mistral series, demonstrating its robustness and generalizability. This confirms that UniMoT can be successfully applied to other SOTA LLMs.

**Comparison of models with comparable sizes.**

We provide a detailed performance comparison between UniMoT and MolCA using models of comparable sizes, as shown in Table 2 of the *global rebuttal PDF*. UniMoT consistently outperforms MolCA across the Galactica-125M, Galactica-1.3B, and LLaMA-2-7B backbones, demonstrating the effectiveness of our proposed UniMoT model.

**Ablation study on codebook size.**

The choice of 2048 for the molecule codebook size is based on a balance between **model complexity** and **performance**. A larger codebook could potentially capture more subtle interactions between molecules and text. However, there may be some codes that are not often used on large codebooks. A smaller codebook may result in nearby embeddings being assigned the same code, which reduces the granularity of the representation.

We conducted experiments with different codebook sizes and report the performance of the molecule captioning task on the Pubchem dataset. The performance with different codebook sizes is shown in Table 5 of the *global rebuttal PDF*. The results demonstrate that the codebook size of 2048 consistently provides the best performance.

**Ablation study on query size.**

We also conducted an ablation study to evaluate the performance of UniMoT with different query sizes, as presented in Table 6 of the *global rebuttal PDF*. The results show that increasing the query size improves performance, with the best results observed at a query size of 32. However, since a query size of 32 requires significantly more training time and memory, we still opt to use a query size of 16.

---

### Decision · Program_Chairs · 2024-09-25

**Decision:**

Reject

**Comment:**

This work presents a new molecule LLM that uses a pretrained tokenizer to replace the projection layer. The tokenizer consists a Q-Former and a VQ module, which are trained with consistency loss. The model is evaluated on molecular understanding and generation tasks. Reviewers have concerns about the novelty (given the existing works on Q-former and VQ-VAE) and dataset usage.